

# Cross-validation of GPS tomography models and methodological improvements using CORS network

Hugues Brenot[1*], Witold Rohm[2], Michal Kačmařík[3], Gregor Möller[4], André Sá[5], Damian Tondás[2], Lukas Rapant[6], Riccardo Biondi[7], Toby Manning[8] and Cédric Champollion[9]

[1]Royal Belgium Institute for Space Aeronomy, Brussels, Belgium.
[2]Wroclaw University of Environmental and Life Sciences, Wrocław, Poland.
[3]Department of Geoinformatics, VŠB-Technical University of Ostrava, Ostrava, The Czech Republic.
[4]Vienna University of Technology, Vienna, Austria.
[5]University of Beira Interior, Covilhã, Portugal.
[6]IT4Innovations, VŠB-Technical University of Ostrava, Ostrava, The Czech Republic.
[7]Università degli Studi di Padova, Padova, Italy.
[8]Royal Melbourne Institute of Technology University, Melbourne, Australia.
[9]Géosciences Montpellier, CNRS, Univ. Montpellier, UA, Montpellier, France.

* Hugues Brenot is also members of the Solar-Terrestrial Centre of Excellence (STCE)

*Correspondence to*: H. Brenot (brenot@aeronomie.be)

**Abstract**

Using data from the Continuously Operating Reference Stations (CORS), recorded in March 2010 during severe weather in the Victoria State, in southern Australia, sensitivity and statistical results of GPS tomography retrievals (water vapour density and wet refractivity) from 5 models have been tested and verified - considering independent observations from 25  radiosonde and radio occultation profiles. The impact of initial conditions, associated with different time-convergence of tomography inversion, can reduce the normalised RMS of the tomography solution with respect to radiosonde estimates by a multiple (up to more than 3). Thereby it is illustrated that the quality of the apriori data in combination with iterative processing is critical, independently of the choice of the tomography model. However, the use of data stacking and pseudo-slant observations can significantly improve the quality of the retrievals, due to a better geometrical distribution and a better 30  coverage of mid- and low-tropospheric parts. Besides, the impact of the uncertainty of GPS observations has been investigated, showing the interest of using several sets of data input to evaluate tomography retrievals in comparison to independent external measurements, and to estimate simultaneously the quality of NWP outputs. Finally, a comparison of our multi-model tomography with numerical weather prediction from ACCESS-A model shows the relevant use of tomography retrieval to improve the understanding of such severe weather conditions, especially about the initiation of the 35  deep convection.



# 1 Potential of GPS tomography for meteorological applications

The GPS tomography considers the use of slant integrated estimates, wet delays or corresponding water vapour contents estimated from the data records of ground-based GPS stations, to retrieve respectively the 3D-field of the wet refractivity or the water vapour density, as introduced by Flores et al. (2000) and Seko et al. (2000). Comparisons of tomography retrievals

with other techniques (water vapour radiometer, radiosonde, raman lidar and atmospheric emitted radiance interferometer) and with numerical weather models have shown relevant results and encouraging understanding of the meteorological situations (Elgered et al., 1991; Gradinarsky, 2002; Gradinarsky and Jarlemark, 2004; Champollion et al., 2005; Bastin et al., 2005, 2007; Troller et al., 2006; Nilsson et al., 2006, 2007; Champollion et al., 2009; Bender et al., 2011a; Perler et al., 2011; Notarpietro et al., 2011; Van Baelen et al., 2011; Brenot et al., 2014). Depending on the resolution of the network of stations,

for the best scenario GPS tomography can adjust these fields with a horizontal resolution of few kilometres, a vertical resolution of ~500 m in the lower troposphere and ~2 km in the upper troposphere, with a time resolution of 15 min. However, to obtain this remarkable geometrical resolution, data from a dense homogeneously distributed network of GPS stations (e.g. 5–25 km spacing) are required, ideally with stations at different altitudes (range of a few hundred meters) according to the orography (Brenot et al., 2012, 2014; Rohm and Bosy, 2009, 2011).

The description of the humidity field, especially the flux of water vapour in the lower atmosphere, is critical for the understanding of convective meteorological events by forecasters and the quantitative prediction of precipitations (Ducrocq et al., 2002). Looking at GPS tomography, the limitation and reliability of retrievals can be linked to two main hurdles. The first one is the limitation of geometrical resolution of the grid adjusted by the inversion process. This limitation is directly linked to the resolution of the network of stations considered. Even Bender et al. (2011b) have shown minor expected

improvements by combining GPS, GLONASS and Galileo observations, this limitation is still present, especially for the volume pixels (so called voxels) of the lower troposphere. The second one, deeply linked to the lack of geometrical distribution and representativeness of retrievals, concerns the convergence of the inversion process to a reliable solution. Improvements of the tomographic method, with the aim to obtain the best convergence of retrievals, have been developed by Champollion et al. (2009) using singular value decomposition combined with a Kalman filter, by Bender et al. (2011a) with

algebraic reconstruction, by Perler et al. (2011) with the use of new parametrised approaches, and more recently by Rohm (2013) with an unconstrained approach and the use of a robust Kalman filter (Rohm et al., 2014) or by Zhao et al. (2017) considering various input observation weighting schemes. Note that for all these methods, the observation-apriori weighting scheme affects the inversion process, and the redundancy or the conflict of information from GPS slant observations crossing the same voxel, are critical for the good achievement of tomography technique.

Taking into account the limitation induced by the geometrical distribution of GPS observations and the issue of the convergence process of the tomographic inversion, this study aims to operate sensitivity tests and methodological improvements using 5 independent tomography models. The same dataset of GPS slant observations has been fed into 5 models and statistical scores (bias, standard deviation and root mean square error) are shown in reference to independent





observations from radiosonde and radio-occultation profiles. After having briefly mentioned, in the previous paragraph, the state of the art of GPS tomography and the limitation this technique is facing, the following section will describe the meteorological situation selected for this study and a description of the GPS data and independent observations considered.

In section 3, the 5 tomography models of this study will be characterised (i.e. the setting considered to prepare the calculations, the algorithm of convergence processed, the approach used to constrain apriori condition and the weight attributed to the data, and the intrinsic specificity of each model and their quality control). The section 4 will present the sensitivity tests and the methodological improvements suggested to face the limitation of GPS tomography. Considerations about the precision of the tomographic solutions, the impact of the apriori condition in the convergence process, and methods for improving the geometrical representativeness of retrievals will be presented. Finally, this study will be concluded about

the interest of using several tomography models and the possible future benefit for operational meteorological applications in case of severe weather and deep convection.

## 2 Severe weather condition in Australian Victoria region and observations studied

Although the pattern of Australia's rainfall accumulation is one of the lowest of the Earth, mainly due to the fact that deserts cover slightly half of the surface of this country, heavy rainfall can take place seasonally, i.e. in the northern equatorial

region with tropical climate and in the southern sub-tropical region with temperate climate. This study focusses on torrential rainfall which happens occasionally during autumn season, in the southeaster Australian Victoria region.

### 2.1 Meteorological situation of the March 2010 Melbourne storms

Heavy rainfall affected the state of Victoria from the 6[th] to the 8[th] of March 2010 and caused consequent flash-floods that disrupted the life of the population of the greater Melbourne region, with several injuries and damages (Choy et al., 2011).

This event is probably the most severe one of the last decade for this region (Jenkins and Lillebuen, 2010). Such event, acting as a Mesoscale Convective System (MCS) with generally a horizontal extension over 10 km, a long life-time (sometimes few hours), and a strong vertical vorticity, largely caused by wind shear, is called a supercell storm. The mean monthly precipitation of Melbourne (50 mm) felt in only one day (61 mm on the 6[th] of March 2010), with fall of hailstones with a size up to 5 cm, as reported by the Australian Bureau of Meteorology (Le Marshall et al., 2010).

Considering estimations of cloud top altitudes from GOME-2 (Global Ozone Monitoring Experiment–2) and OMI (Ozone Monitoring Instrument) sensors, a vertical extension at least higher than 10 km induced by a supercell storm, is observed on Fig. 1 close to Melbourne. Cloud top altitudes have been evaluated using cloud top pressure retrievals (Lutz et al., 2016; Vasilkov et al., 2008) combined to the assumption of an atmosphere in hydrostatic equilibrium. This application of the hypsometric equations requires inputs of ground pressure and mean temperature of the column of air for each footprint

considered. These two parameters have been obtained from the outputs of the analysis of ACCESS-A (Australian Community Climate and Earth-System Simulator, with a focus on Australian area), the operational Numerical Weather





Prediction (NWP) of the Australian Bureau of Meteorology (Puri et al., 2013). GOME-2 and OMI UV-visible instruments are on board of polar orbiting satellite platforms (MetOp-A and Aura respectively) and generally provide, for latitudes under 50° (North or South), one measurement per day and location. Fig. 1a and 1b show the cloud top altitudes, obtained by GOME-2 and OMI during overpass over Melbourne at 23:56 UTC (on 2010/03/05) and 04:07 UTC (on 2010/03/06), respectively. On Fig. 1a, the MCS (with vertical extension over 9 km) covers an area of 80×80 km² (the size of a pixel from GOME-2 instrument is 80×40 km²), and the area with stratiform precipitation (vertical extension over 6 km) is about 160×160 km².

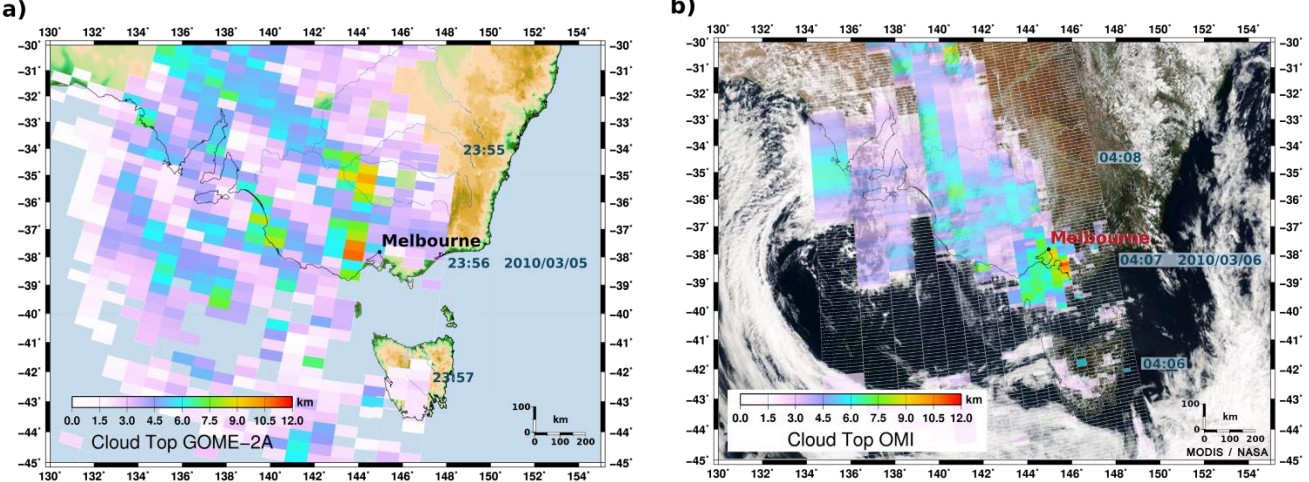

**Figure 1: Observations of Cloud Top altitudes from a) GOME-2 at 23:56 UTC on 2010/03/05, and b) OMI satellite's sensors at 04:07 UTC on 2010/03/06 (background from MODIS/Aqua).**

On Fig 1b., convective precipitation with high vertical extension concerns an area of about 24×84 km² (a pixel from OMI is 24×13 km²), and stratiform precipitation in an area of 168×195 km². The background of Fig. 1b is an image (visible channel) from MODIS (Moderate Resolution Imaging Spectroradiometer). This imager, on board Aqua satellite, simultaneously looks at the same area of the Earth as OMI/Aura. The dimension of the MCS detected by GOME-2 and OMI are respectively confirmed by heavy and moderate precipitation recorded by the weather radar of Melbourne; see Choy et al. (2011) and Manning et al. (2012) for more details about this meteorological situation.

A typical configuration of the atmosphere, with instability in the troposphere (e.g. triggered by the topography or due to low level convergence) associated with the opposition of dry and moist air, has been required to initiate deep convection during the 6-8 March superstorm of greater Melbourne (Choy et al., 2011). At the end of the day on 5 March 2010 (UTC time), lifting of moist coming from the south and southeast side of Melbourne and warm dry air coming westwards of Melbourne triggered the convection. The water vapour content of the troposphere is a key parameter to initiate and keep running deep convection. Its monitoring is critical to evaluate the severity of severe weather events (nowcasting) and to forecast meteorological situations. Figure 2 shows the 2D field of Integrated Water Vapour (IWV) from ACCESS-A NWP during the

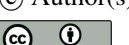



supercell storm of Melbourne in March 2010. Even the horizontal resolution of ACCESS-A outputs is 12 km, the weather prediction used in this study, was not sufficient to properly forecast this storm and the creation of such a supercell. Strong differences with the IWV fields from GPS are observed in Fig. 2. The flux of water vapour from the south of Melbourne is indicated by horizontal IWV gradients, illustrating the critical condition that took place in the region surrounding Melbourne,

which kept going the convective process inside the supercell. Monitoring by GPS technique using dense network can be essential for improving our capability to better understand and forecast such events. For this purpose, this study focusses on testing of 5 different tomography softwares for estimating reliable tomography retrievals.

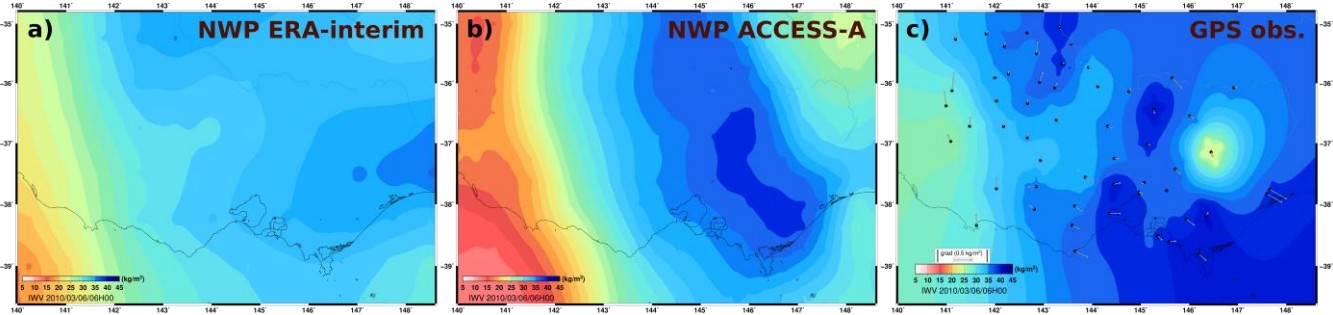

**Figure 2: 2D fields of IWV from a) ERA-interim NWP, b) ACCESS-A NWP, and c) GPS observations at 06:00 UTC on 2010/03/06**
**(the black circles are the locations of GPS stations and the grey arrows show the horizontal IWV gradients).**

## 2.2 Inputs for GPS tomography using CORS network

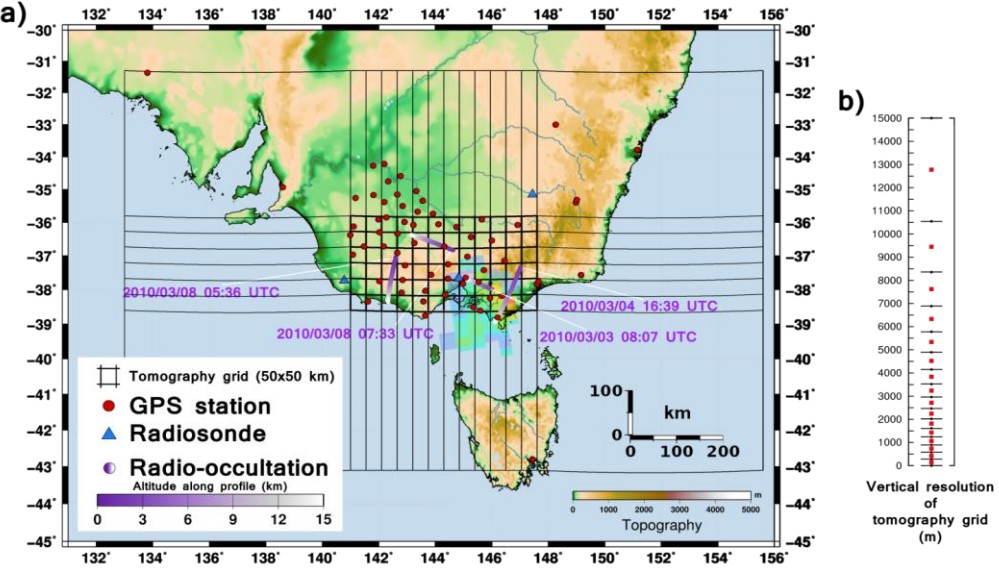

**Figure 3: a) GPS stations (red circles), Radiosonde sites (blue triangles), Radio-occultation profiles (white-purple lines), OMI cloud**
**top altitude at 04:07 on 2010/03/06 (purple-blue-orange pattern; c.f. Fig. 1b) and the tomography grid (inner grid in thick black lines and outer grid in thin black lines); b) Red square represents altitudes above the sea level of the centres of each voxel of the tomography grid.**



During the Melbourne storm in March 2010, 69 stations of the Victorian GPS CORS network (see Fig. 3a for the locations of these stations) were recording signals emitted from 31 GPS satellites. Data has been processed by the Royal Melbourne Institute of Technology University using double difference technique with Bernese 5.0 software (Hugentobler et al., 2007). Thereby, Zenith Total Delays of the neutral atmosphere (ZTDs) and horizontal delay gradients $\vec{G} = \begin{pmatrix} G_{NS} \\ G_{EW} \end{pmatrix}$ have been

retrieved every 30 minutes, covering the period from the 3$^{rd}$ to the 10$^{th}$ of March 2010.

For GPS tomography and to assess the quality of the derived 3D fields of wet refractivity and water vapour density, the GPS tropospheric estimates have been converted into Slant Wet Delays (SWDs) and Slant Integrated Water Vapour contents (SIWVs), respectively. Estimates of ground pressure and temperature are required for each station of the GPS CORS network. These parameters have been obtained using outputs from ACCESS-A weather model. A horizontal spatial

interpolation to the position of each GPS site, and a vertical interpolation for the temperature and the assumption of an atmosphere in hydrostatic equilibrium, have been considered to obtain pressure and temperature for each station. A time interpolation has also been applied from 6 hours of weather model to a 30-minute time resolution of GPS tropospheric parameters. The formula of Saastamoinen (1972) has been used to convert the pressure data into Zenith Hydrostatic Delay (ZHD). The Zenith Wet Delay (ZWD) is obtained simply subtracting ZHD from ZTD estimates (i.e. ZWD = ZTD – ZHD).

Thereby, other contributions, like hydrometeor delays, has not been considered. Using the field of ground pressure from ACCESS-A, the hydrostatic part of the azimuthal contribution of GPS gradient to delays to slant delays, has been removed to keep only the wet contribution (see Champollion et al., 2004; Brenot et al., 2014).

The elevation ($\epsilon$) and azimuth ($\alpha$) dependency of the SWD is described by an isotropic ($L_{az}^{wet}$) and anisotropic ($L_{az}^{wet}$) component.

$$SWD = L_{sym}^{wet}(\epsilon) + L_{az}^{wet}(\epsilon, \alpha) \tag{1}$$

The isotropic SWD ($L_{az}^{wet}$) is obtained by mapping of the ZWD using a wet mapping function (in our case GMF from Boehm et al., 2006) in direction of the GPS satellite in view above 5° elevation.

$$L_{sym}^{wet}(\epsilon) = ZWD \cdot mf_{sym}^{wet}(\epsilon) \tag{2}$$

The first order wet anisotropic contribution ($L_{az}^{wet}$) is formulated in Eq. 3.

$$L_{az}^{wet}(\epsilon, \alpha) = mf_{az}^{wet}(\epsilon, C) \cdot (G_{NS}^{wet} \cdot \cos(\alpha) + G_{EW}^{wet} \cdot \sin(\alpha)) \tag{3}$$

with gradient mapping function ($mf_{az}^{wet} = 1/(\sin\epsilon\tan\epsilon + C)$) which depends on satellite's elevation and on a constant $C = 0.0032$ (Herring, 1992; Chen and Herring, 1997), and the gradient wet components ($G_{NS}^{wet}$, $G_{EW}^{wet}$) in north-south and east-west direction, which are multiplied with the cosine or sine of azimuth ($\alpha$), respectively. Note that one-way residual has not been considered in $L_{az}^{wet}$ and SWD retrievals.



The gradient components ($G_{NS}$, $G_{EW}$) retrieved by GPS technique describe the total asymmetric effect. This means there is no distinction between wet and hydrostatic gradients (Chen and Herring, 1997; Flores et al., 2000). However, the wet gradient component can be expressed by the difference of the hydrostatic to the total component as follows:

$$\begin{pmatrix} G_{NS}^{\text{wet}} \\ G_{EW}^{\text{wet}} \end{pmatrix} = \begin{pmatrix} G_{NS} \\ G_{EW} \end{pmatrix} - \begin{pmatrix} G_{NS}^{\text{hydrostatic}} \\ G_{EW}^{\text{hydrostatic}} \end{pmatrix} \tag{4}$$

To obtain the hydrostatic gradient components ($G_{NS}^{\text{hydrostatic}}$ and $G_{EW}^{\text{hydrostatic}}$), a characterisation of the surface pressure field around each GPS station is required. In that case, the hydrostatic gradient can be established by fitting a plane through the pressure measurements (Champollion et al., 2004; Brenot et al., 2014). From the pressure field near a GPS site, the spatial variations of the hydrostatic delay per unit of distance (km) in the north-south ($Z_{NS}^{\text{hydrostatic}}$) and east-west ($Z_{EW}^{\text{hydrostatic}}$) directions can be calculated. Surface pressure measurements around all GPS stations of CORS network were not available during the supercell storm of 6-8 march 2010. For this reason, the pressure field of ACCESS-A NWP has been considered. Assuming an exponential law in the hydrostatic refractivity and considering the scale height of the gradients in the hydrostatic delays set to $H = 13$ km (as suggested by Chen and Herring, 1997), the spatial variations of the hydrostatic delay can be converted in hydrostatic gradients (Elòsegui et al., 1999; Ruffini et al., 1999; Flores et al., 2000):

$$\begin{pmatrix} G_{NS}^{\text{wet}} \\ G_{EW}^{\text{wet}} \end{pmatrix} = H \cdot \begin{pmatrix} Z_{NS}^{\text{hydrostatic}} \\ Z_{EW}^{\text{hydrostatic}} \end{pmatrix} \tag{5}$$

The resulting slant wet delays $\left( \text{SWD} = \int N_w \, ds \right)$ are used as input for GPS tomography to retrieve 3D fields of wet refractivity ($N_w$). The second type of slant measurements used, is water vapour slant total columns density, also called slant integrated water vapour contents (SIWVs), which can be simply obtained from SWDs by multiplication with factor κ (Eq. 6). Then GPS tomography retrieves the 3D field of water vapour density ($\rho_{wv}$).

$$\text{SIWV} = \kappa \cdot \text{SWD} = \int \rho_{wv} \, ds \tag{6}$$

The expression of κ factor (Askne and Nordius, 1987) requires an estimation of the mean temperature of the column of air ($T_m$) above the GPS station. Therefore, temperature profiles from ACCESS-A have been interpolated in space and time. The obtained κ factors were applied in Eq. 6 for the conversion of SWDs into SIWVs.

From the 3[rd] to the 10[th] of March 2010 (288 epochs, 30-minute time resolution) in total 197429 pairs of SWDs/SIWVs have been retrieved (about 686 pairs at each time). These observations are the basis input data for our tomography and sensitivity tests.



### 2.3 Independent observations of water vapour density and wet refractivity profiles

During the period of the heavy rainfall events in the beginning of March 2010, two independent sets of $N_w$ and $\rho_{wv}$ profiles, derived from radiosonde (RS) and GPS radio-occultation (RO) measurements, have been considered to quantify the sensitivity of the tomography tests proceeded and to verify/validate the possible improvements suggested.

### 2.3.1 Profiles from radiosonde

The most common and accurate yet quite expensive technique to measure the vertical profile of temperature, pressure, dew point temperature, wind speed and direction, is a sensor attached to an automatic radio-sounding balloon released either every 6, 12 or minimum 24 hours. Usually profiles reach up to 30 km, and have very high vertical resolution, major drawback of such measurements is horizontal movement of balloon as it ascents vertically, due to the winds. The ascend time is around 1 hour. In 2010 in Australia there were 32 radiosonde stations with only one located in Victoria (Melbourne airport - YMML) and other two in the short distance to the Victorian border (Waga Waga - YSWG and Mount Gambier - YMMG), as shown in Fig. 3. Only YMML RS station, located at an altitude of 141 m and position of (37.67°S, 144.83°E), has been considered, with RS data available twice a day (at noon and midnight) for all days used in the study. The radiosonde technique is highly accurate for observations of the troposphere with a 1-2 hPa accuracy for pressure, 0.5 °C accuracy for temperature and 5 % accuracy for relative humidity (Manning, 2013). Following calculations has been made to retrieve wet refractivity and water vapour content: 1) The formula by Sargent (1980) was applied to obtain the relative humidity from the dew point temperature (Lawrence, 2005). 2) The formula by Sonntag (1994) was used to calculate partial pressure of water vapour $p_w$ from $RH$ and $T$. 3) The atmospheric state parameters $p_w$, $p$ and $T$ were used to calculate wet refractivity $N_w$ and water density profiles $\rho_{wv}$, using equations proposed by Davis et al., (1985):

$$N = k_1 R_d \rho + k'_2 \frac{p_w}{T} + k_3 \frac{p_w}{T^2} \tag{7}$$

where $k_1, k'_2$ and $k_3$ denote empirically-derived refractivity coefficients, $R_d$ is a gas constant for dry air and $\rho$ is atmospheric density.

### 2.3.2 Profiles from radio-occultation technique

The RO profiles used in this study are the wet profiles acquired by the Constellation Observing System for Meteorology, Ionosphere, and Climate (COSMIC) and re-processed in 2013 at COSMIC Data Analysis Archive Center (CDAAC) reported versus the geometric height above the mean sea level. The wet profile (wetPrf), is an interpolated product with a vertical resolution of 100 m obtained using a one dimensional variational (1DVar) technique together with European Centre Medium Weather Forecast (ECMWF) low resolution analysis data and it contains latitude and longitude of the perigee point, pressure, Temperature ($T$), water vapour pressure ($p_w$), refractivity and a mean sea level altitude of the perigee point. The atmospheric parameters have been computed using a 1DVar approach, where refractivity is weighted in the way that the



temperature is basically the same as the dry temperature in regions where the water vapour is insignificant (Biondi et al., 2011). Analyses on specific humidity estimated by GPS RO (Pincus et al., 2017) have shown consistency within 0.1–0.3 g/kg in the median with the Global Climate Observing System (GCOS) Reference Upper-Air Network (GRUAN) radiosonde data and with theoretical studies of accuracy (e.g. Kursinski and Gebhardt 2014; Ladstädter et al. 2015). Kuo et al., (2005)

5    states that RO performs better than radiosondes between 5 and 25 km and in Australia region ECMWF–RO is not of better performance while the ECMWF-RS was shown to be biased between 5 and 15km.

We have converted the $p_w$ in $\rho_{wv}$ by using the gas equation:

$$\rho_{wv} = \frac{p_w}{461.525 * T} * 1000 \qquad (8)$$

for the four RO reported in Tab. 1 and shown in Fig. 3. All the profiles were acquired by the COSMIC satellites.

15    **Table 1: List of Radio-occultations.**

| Mean times of measurement | Cosmic-GPS couple | Lower position | Top of tomo. grid | Higher position |
|---|---|---|---|---|
| *2010/03/03* 08:07 UTC | C006-G09 | 37.99°S 145.98°E 2300 m | 37.71°S 145.17°E 12800 m | 37.66°S 144.62°E 39900 m |
| *2010/03/04* 16:39 UTC | C004-G12 | 37.33°S 147.03°E 1900 m | 38.81°S 146.21°E 12800 m | 39.65°S 145.53°E 39900 m |
| *2010/03/08* 05:36 UTC | C003-G01 | 37.04°S 142.60°E 1200 m | 38.25°S 142.33°E 12800 m | 38.86°S 142.35°E 39900 m |
| *2010/03/08* 07:33 UTC | C006-G27 | 36.83°S 144.63°E 1300 m | 36.41°S 143.26°E 12800 m | 36.32°S 142.52°E 39900 m |

**3 Selection of tomography models**

GPS tomography uses slant integrated measurements (SWDs / SIWVs) inside a defined volume to retrieve $N_w$ or $\rho_{wv}$ estimates for each voxel $N_{pqr}$, as illustrated in Fig. 4. For highest consistency, the same GPS data set and the same 3D grid

20    was used for each of the 5 tomography models considered in this study.





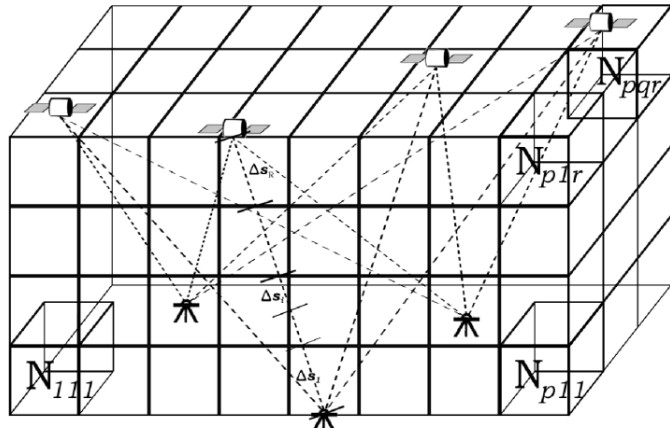

**Figure 4: Discretisation of the space by a tomographic grid where unknowns (wet refractivity or water vapour densities) are the values ($N_{xyz}$) of each voxel. Dash lines are rays of propagation of GPS signals from satellites to ground stations.**

For each GPS station within the voxel model, the integrated slants observations $SLANT_{GPS}$ (in our case the SWDs or

SIWVs) were converted into 3D fields of $N_w$ or $\rho_{wv}$., using Eq. (9):

$$SLANT_{GPS}\ (P, \alpha, \epsilon)\ =\ \sum_{i=1}^{NB_{vx}} N_{x_i y_i z_i}\ \Delta s_i \tag{9}$$

where P is the position of a GPS station (latitude, longitude, altitude), $\epsilon$ is the elevation and $\alpha$ the azimuth angles of a GPS

satellite, $NB_{vx}$ is the number of voxels crossed by each observation and $\Delta s_i$ is the ray length in each voxel $i$ ($x_i \in [1, p]$, $y_i \in$

[1, $q$], $z_i \in [1, r]$). Hereby, axes x, y, z are associated respectively to longitude, latitude and altitude. For the tomographic

grid, the horizontal size of the voxels in the inner grid is 0.5° ~ 50 km (thick black lines of Fig. 3a). 12 voxels are considered

longitudinally ($p$ = 12) and 6 voxels latitudinally ($q$ = 6). All around this inner grid, voxels of the outer grid are considered

inside a band with a width of about 400 km (thin black lines of Fig. 3a). The outer grid is used to keep considering in our

calculations $SLANT_{GPS}$ of stations located close to the edge of the inner grid, and for which the ray does not reach the top of

the tomographic grid inside the inner grid. To deal with the topography under the tomographic grid, note that the lower layer

of the grid is at the altitude of the sea level (altitude = 0 m a.s.l.). The voxels for which there is no GPS station inside or

under, are simply not retrieved by tomography models. Section 4 and Table 3 will discuss about the percentage of voxels

retrieved in our calculations, according of different types of calculations and settings.

Assuming the hypothesis of straight rays (the bending effect which can affect observations under 15° elevations (Möller,

2017) has not been considered in our study), the inversion problem becomes linear. The formulation of the linear inverse

problem reads:

$$\mathbf{d} =\ \mathbf{Gm} +\ \mathbf{C_D} \tag{10}$$



where **d** represents the SWD/SIWV observations which are connected to the model **m** of wet refractivity / water vapour density ($N_w$ / $\rho_{wv}$ that we want to retrieve) by the geometrical matrix **G** and the observation errors, defined by the covariance operator $\mathbf{C_D}$. Positions of stations and satellites which characterise the geometrical matrix (positions expressed in Cartesian coordinates) and SWD/SIWV observations with the associated errors ($\Delta$SWD/$\Delta$SIWV expressed in $\mathbf{C_D}$) are the inputs of this inverse problem. The solution **m** does not exist (GPS tomography is an ill-posed problem). However the alternative is to find **m'** which minimises the $L_2$-norm ($\xi$ is the minimisation factor):

$$\|\mathbf{d} - \mathbf{Gm'}\|_{L_2} < \xi \tag{11}$$

Different standard techniques exist for solving such linear inverse problems: e.g. Singular Value Decomposition (SVD), Truncated Singular Value Decomposition (TSVD), Weighted and damped least-squares solution, Kalman-filter, Algebraic Reconstruction Techniques (ART), Conjugate gradient method, Tikhonov-regularisation, Stacking or double-difference– spline interpolation. The next sections present the specificity of the 5 tomography models used in this study. Note that the same apriori condition $\mathbf{m_0}$ has been considered by these models. It is based on 6 hours NWP outputs from ACCESS-A, interpolated to the tomography grid of Fig. 3. See Puri et al. (2013) for more details about the observations assimilated in ACCESS-A using a four-dimensional observation variational assimilation method (4DVAR). In 2010, the radio occultation observations were not assimilated in ACCESS-A system.

### 3.1 3D tomography using SVD at BIRA

The tomography model used at the Royal Belgian Institute for Space Aeronomy (BIRA-IASB) is an adaptation of LOFTT$_K$ software, developed by Champollion et al. (2005). In its current version this software uses weighted and damped least-squares inversion with SVD. The inversion by least-squares adjustment resolves the $N_w$ or $\rho_{wv}$ model minimising the misfit function $\mathbf{\chi^2}$ considering $L_2$-norm (Tarantola, 2005):

$$\mathbf{\chi^2} = (\mathbf{Gm} - \mathbf{d})^\mathbf{T} \, \mathbf{C_D^{-1}} \, (\mathbf{Gm} - \mathbf{d}) + (\mathbf{m} - \mathbf{m_0})^\mathbf{T} \, \mathbf{C_M^{-1}} \, (\mathbf{m} - \mathbf{m_0}) \tag{12}$$

where $\mathbf{C_M}$ is the covariance operator associated to the apriori model $\mathbf{m_0}$ of $N_w$ or $\rho_{wv}$. The inversion by least-squares is optimal when the errors follow a Gaussian law (hypothesis we can assume). Note that if this is not the case, the least-squares inversion gives an approximate solution. The uniqueness of the solution depends on the geometry of the network and the number of observations. For GPS tomography, the geometry of network is unfavourable because the atmosphere is only scanned from the surface. All the voxels are not lit by GPS rays (under-determined problem) but the information brought by the set of rays is redundant for some voxels (over-determined problem). We speak then about a mix-determined or ill-posed inverse problem. In this case, the inverse of the geometrical matrix (or other matrix linked to it; e.g. the inverse of **A** matrix described in Eq. 14) does not exist. An approximate solution for $\mathbf{A^{-1}}$ can be obtained from the SVD technique and the generalised inverse $\mathbf{A^{-g}}$. The use of an apriori model $\mathbf{m_0}$ in least-squares process is also a good way to optimise the adjustment of the solution **m**. To express the associated covariance operator $\mathbf{C_M}$ to apriori model, external measurements





(like radiosonde profiles or climate weather models) can be considered. In this tomography software, $\mathbf{m_0}$ itself (outputs from ACCESS-A) and a damping coefficient ($\boldsymbol{\delta_M}$ = 0.9) have been applied to describe the covariance operator of the apriori model ($\mathbf{C_M} = \boldsymbol{\delta_M}\,\mathbf{m_0}$). The weight is directly linked to the values of the apriori model $\mathbf{m_0}$. The closer $\boldsymbol{\delta_M}$ is to 1, the more difficult it is for solution $\mathbf{m}$ to differ from $\mathbf{m_0}$. Note that in the same way, the covariance operator $\mathbf{C_D}$ of data observations

has been estimated by ($\mathbf{C_D} = \boldsymbol{\delta_D}\,\mathbf{d}$) with a damping coefficient ($\boldsymbol{\delta_D}$ = 0.1). The higher $\boldsymbol{\delta_D}$ is, the higher is the freedom applied in the adjustment of solution $\mathbf{m}$.

The least-squares solution $\mathbf{m}$ of ($N_w$ or $\rho_{wv}$) has been retrieved using formulation of Eq. (11):

$$\mathbf{m} = \mathbf{m_0} + \mathbf{C_M}\mathbf{G^T}\mathbf{A^{-g}}\,(\mathbf{d} - \mathbf{Gm_0}) \tag{13}$$

with

$$\mathbf{A} = \mathbf{GC_M}\mathbf{G^T} + \mathbf{C_D} \tag{14}$$

Note that an equivalent expression: $\mathbf{m} = \mathbf{m_0} + \left(\mathbf{G^T}\mathbf{C_D^{-1}}\mathbf{G} + \mathbf{C_M^{-1}}\right)^{-g}\mathbf{G^T}\mathbf{C_D^{-1}}\,(\mathbf{d} - \mathbf{Gm_0})$, for Eq. (13) can also be found in the literature (Tarantola, 2005; p. 36, Eq. 1,106).

The SVD concerns a factorisation of $\mathbf{A}$ matrix (called covariance square matrix and is linked to $\mathbf{G}$, $\mathbf{C_M}$ and $\mathbf{C_D}$), as shown in

Eq. (15):

$$\mathbf{A} = \mathbf{U}\boldsymbol{\Lambda}\mathbf{V^T} \tag{15}$$

where $\mathbf{U}$ is the orthonormal matrix (data space), $\boldsymbol{\Lambda}$ the diagonal singular values matrix, and $\mathbf{V}$ the orthonormal matrix (model parameter space). $\mathbf{A^{-1}}$ can not be obtained and for this reason, the generalised inverse $\mathbf{A^{-g}}$ is finally used, as formulated in

Eq. (16) by:

$$\mathbf{A^{-g}} = \mathbf{V}\boldsymbol{\Lambda^{-1}}\mathbf{U^T} \tag{16}$$

This methodology provides 6 quality control parameters: 1) the accumulated distance of each ray inside each voxel (geom[ray]), 2) the diagonal elements (diag[resol]) of resolution matrix $\mathbf{R}$, 3) the trace (trace[resol]) of $\mathbf{R}$, 4) the diagonal element (diag[cova]) of covariance square matrix $\mathbf{A}$, 5) the spread factor (spread[f]) which is linked to the distance to the next voxel and

the resolution matrix, 6) the condition number (cond[num]) that is the ratio between the maximum and the minimum singular values (diagonal elements of matrix $\boldsymbol{\Lambda}$). The resolution matrix is defined by: $\mathbf{R} = \mathbf{C_M}\mathbf{G^T}\mathbf{A^{-g}}\mathbf{G}$. Quality controls diag[resol] and diag[cova] are generally low (under note that the closer diag[resol] and diag[cova] are to 1, the closer is trace[resol] to the number of observations, the best is the solution. Concerning cond[num], because GPS tomography is an ill-posed problem, this value of the singular values can be extremely high (from $10^{10}$ to $10^{12}$ for the maximum values, and from $10^8$ to $10^9$ for the minimum).

To facilitate the visualisation of this quality control, the ratio between maximum and minimum singular values (cond[num] from 50 to 2000) can show the quality of retrievals. If cond[num] is low (under 100), the quality of retrievals can be considered good.





## 3.2 3D tomography using SVD and a robust Kalman filter at WUELS

In TOMO2, the tomography software from Wroclaw University of Environmental and Life Sciences (WUELS), robust Kalman filter techniques have been used to determine $N_w$ or $\rho_{wv}$ values, for more details the reader is referred to Rohm et al., (2014). This algorithm estimates the system state, based on the previous state and the correction to the measurement. In case of $N_w$, determining the future state $N_{w_{k+1}}$ is described by formula (Eq. 17):

$$N_{w_{k+1}} = \Phi_k \cdot N_{w_k} + \omega_k \tag{17}$$

where $\Phi_k$ is the system change matrix, and $\omega_k$ contains the system noise with expected value of $E(\omega_k) = 0$, and covariance $E(\omega_k \omega_k^T) = Q_k$, which is called the dynamic noise matrix. The linear observation model is expressed by:

$$SWD_k = A_k \cdot N_{w_k} + \vartheta_k \tag{18}$$

where $SWD_k$ are assumed to be independent slant observations of GPS signal delay, $A_k$ is a design matrix containing $\Delta s_i$ for each $SWD_k$, and $\vartheta_k$ denotes measurement noise with expected value $E(\vartheta_k) = 0$ and covariance $E(\vartheta_k \vartheta_k^T) = R$. The Kalman's robust filter allows to reduce weights for outliers. The covariance matrix of the measurement noise then takes the form $E(\vartheta_k \vartheta_k^T) = R^R$. The form of the variance covariance matrix can also be determined from the previous state based on the following relationship:

$$P_{N_{w_k}}(-) = \Phi_k \cdot P_{N_{w_{k-1}}}(+) \Phi_k^T + Q_{N_{w_{k-1}}} \tag{19}$$

where $N_{w_k}(-)$ is the future state of the system, and $N_{w_{k-1}}(+)$ describes the previous system state for wet refraction, $P_{N_{w_k}}(-)$ is the predicted matrix variance-covariance, whereas $P_{N_{w_{k-1}}}(+)$ is the variance-covariance of the previous system state. As described above, the use of a robust Kalman filter is associated with outliers detection:

$$r_k = A_k \cdot N_{w_k}(-) - SWD_k \tag{20}$$

where $r_k$ is the residual value of observation. They are the basis for assigning new values in the covariance matrix $R^R$:

$$R^R = (diag(p_1, p_2, \dots, p_m))^{-1} \tag{21}$$

where $p_i$ values are determined on the basis of residual values $r_i$:

$$|r_i| \leq \frac{\sigma \cdot c}{\sqrt{p}} \tag{22}$$

where σ is the variance reference value, usually assumed to be 1, c is a scaling parameter whose value is 1.5, and p is the weight of observation. If the condition is fulfilled then $p_i = p$. Otherwise, $p_i$ is calculated from:

$$p_i = \frac{c\sigma\sqrt{p}}{|r_i|} \tag{23}$$




The last phase of the Kalman filter is the state correction. It is based on the so-called Kalman's gain:

$$K_{N_{w_k}} = P_{N_{w_k}}(-)A_k^T(A_k P_{N_{w_k}}(-)A_k^T + R_{N_{w_k}})^{-1} \tag{24}$$

The determination of the state of the system (Eq. 25) together with the variance-covariance matrix (Eq. 26) for the wet refractive index is finally expressed in the form

$$N_{w_k}(+) = N_{w_k}(-) + K_{N_{w_k}}(SWD_k - A_k \cdot N_{w_k}(-)) \tag{25}$$

$$P_{N_{w_k}}(+) = P_{N_{w_k}}(-) + (K_{N_{w_k}} A_k P_{N_{w_k}}(-)) \tag{26}$$

TOMO2 model does not use constraints between voxels, either vertical and horizontal, the solution to (Eq. 24) – definition of Kalman gain is done using Truncated Singular Value Decomposition as discussed in Rohm (2013).

### 3.3 2D tomography using LS adjustment at TUO

2D GPS tomography technique used at VSB-TUO (VSB-Technical University of Ostrava) represents a different approach than typical 3D tomography. It is based on a limited number of GPS reference stations positioned as much as possible in a straight line and builds a single narrow strip of 3D voxels above it. Only slant delays with azimuthal angles similar to orientation of the line are entering the solution. The number of available input slant delays is therefore generally dependent on the line orientation and strongly changes with time due to moving satellite constellation of used GPS. Although $N_w$ or $\rho_{wv}$ values are being estimated in all voxels, the main output of the technique is a vertical profile of $N_w$ or $\rho_{wv}$ in voxels above the middle of the line since these are crossed by the largest number of slant delays. Generally, these values are obtained by solving a linear system (Eq. 27),

$$Ax = b, A \in \mathbb{R}^{m \times n}, \tag{27}$$

where values of $A$ represent a length of a path that signal travels in individual voxels, $x$ represents desired solution (i.e. values of $N_w$ or $\rho_{wv}$) and $b$ represents measured values of each slant delay. These linear systems are rarely well defined (i.e. have same number of equations as variables). More usually, they are either over or underdetermined. This property complicates use of standard techniques of solving systems of linear equations and requires more general approach. One of the possible approaches to solve linear systems is defined by Eq. (28).

$$x = \underset{x_0 \in \mathbb{R}^n}{\text{argmin}}(Ax_0 - b). \tag{28}$$

This form of the problem opens the use of a number of numerical optimisation techniques like least squares or gradient descent. These methods, however, proved to be barely usable due to the high number of local minimums that are present in this problem. Therefore, a more general global optimisation algorithm has been used in a form of simulated annealing (SA), see Zelinka and Skanderova (2012). This method is based on a metallurgical technique involving heating and controlled





cooling of a material to increase the size of its crystals and reduce their defects. Generally, the state of some physical systems, and the function $A$ to be minimised, can be likened to the internal energy of the system in that state. The goal is to move the system from some initial state to a state with the minimum possible energy. At each step, the algorithm finds some neighbouring solutions $x_n$ of the current solution $x_0$, and probabilistically decides between moving the system to state $x_n$ or

staying in state $x_0$. When correctly set, these probabilities lead the system to move to the states with lower energy. This procedure is repeated until the system reaches a state that can be considered good enough for the application, a state that is not changing between iterations or until a given number of iterations has been met. Iterating the algorithm ensure, together with the probabilistic approach for accepting the different states, that it does not stuck in the local optimums. Other bio-inspired metaheuristics like differential evolution or SOMA (see Onwubolu and Babu, 2004) were also considered but none

gave as good results as SA.

We also impose several constraints that should help with the convergence of the method. Values of the optimised parameters are bounded by the initial value of $x_0$. In our experiments, we have chosen to bound the values of $x_0$ to $< \frac{1}{3} ; 3 >$ multiple of the initial value. Also, the residuals of each optimisation step are weighted by the elevation angle of the observation using linearly decreasing weights. For more information about principles of this technique the reader is referred to Kacmarik and

Rapant (2012).

### 3.4 3D tomography using TSVD at TUW

The tomography solution from TU Wien (TUW) is based on the software package for atmospheric tomography (short: ATom). ATom combines weighted least squares techniques with truncated singular value decomposition methods (TSVD) and 2D ray-tracing methods for iterative reconstruction of signal paths and refractivity fields from tropospheric signal

delays. Based on the number of observation types or constraints, the equation system is split into several partial solutions. In case of two subsets (slant delays and apriori information as used within this study), the tomography solution $\mathbf{m}$ and its partial solution $\mathbf{m_D}$ read:

$$\mathbf{m_D} = \mathbf{V\Lambda^{-1}U^T} G_D{}^T C_D^{-1} d \tag{29}$$

where $\mathbf{U}, \mathbf{V}$ and $\mathbf{\Lambda}$ are obtained by singular value decomposition of matrix $G_D{}^T \cdot C_D^{-1} \cdot G_D$. The diagonal elements $\sigma_{D,n}$ of the apriori variance-covariance matrix $\mathbf{C_D}$ are computed as function of elevation: $\sigma_{D,n} = sin^2 \epsilon \cdot \sigma_{ZTD}^2$ where $\sigma_{ZTD}^2 = 2.5mm$ reflects the accuracy of the estimated zenith total delays.

In a second step, solution $\mathbf{m_D}$ is combined with the apriori information $\mathbf{m_0}$ as follows:

$$\mathbf{m} = \mathbf{m_D} + \mathbf{V\Lambda^{-1}U^T} G_0{}^T C_0^{-1}(\mathbf{m_0} - G_0\mathbf{m_D}) \tag{30}$$

where $\mathbf{U}, \mathbf{V}$ and $\mathbf{\Lambda}$ are obtained by singular value decomposition of matrix $G_0{}^T \cdot C_0^{-1} \cdot G_0 + C_{\hat{m}}^{-1}$ with $\mathbf{C_{\hat{m}}} = \mathbf{V\Lambda^{-1}U^T}$ as the variance-covariance matrix of the first partial solution. The diagonal elements $\sigma_{0,n}$ of variance-covariance matrix $\mathbf{C_0}$ were



derived in an initial step from comparisons of the apriori data (ACCESS-A) with radiosonde measurements (altitude dependent weighting).

The benefit of the partial solution is, that $\mathbf{m_D}$ depends solely on the observations ($\boldsymbol{d}$), which allows a proper selection of singular values e.g. by means of L-curve technique (see Möller, 2017). Thereby, less resolved voxels are detected and ejected - individually for each observation type.

Finally, the obtained tomography solution is checked for outliers by analysis of the post-fit residuals. In case of large residuals, outliers are removed and the processing is repeated. Usually 3 to 4 iterations are necessary until the RMS of the residuals converges. The quality of the obtained refractivity fields is assessable by analysis of the normalised misfit function, condition number, residuals and standard deviation of the estimates. The reader is referred to Möller (2017) for more details.

## 3.5 3D tomography using ART at UBI

SEGAL GPS Water Vapour Reconstruction Image Software (SWART) was developed at Space and Earth Analysis Laboratory (SEGAL) in the University of Beira Interior (UBI). SWART uses Algebraic Reconstruction Techniques (ART) to compute $N_w$ or $\rho_{wv}$ tropospheric distribution over a specified area using SWDs or SIWVs, respectively, and plots results as 2D images of horizontal or vertical sections.

Various algebraic iterative methods for reconstruction, i.e. for solving large linear systems (see form presented in Eq. 27), are used in tomography and many other inverse problems (Censor and Elfving, 2002; Jiang and Wang, 2003a, 2003b; Censor et al., 2007; Qu et al., 2009; Bender et al., 2011b), and were implemented in SWART. In this specific case, for the inversion, we used the Simultaneous Algebraic Reconstruction Technique (SART) that was implemented as a Simultaneous Iterative Reconstruction Technique (SIRT).

These SIRT methods are "simultaneous" in the sense that all parameters in vector $x$ are updated at the same time in one iteration. The method can be written in the general form:

$$x^{k+1} = x^k + \lambda_k T A^T M(b - Ax^k), \qquad k = 0,1,2,\ldots \tag{31}$$

where $x^k$ denotes the current iteration vector, $x^{k+1}$ is the new iteration vector, $\lambda_k$ is a relaxation parameter, and the matrices $M$ and $T$ are symmetric positive definite. Different methods depend on the choice of these matrices. The iterates of the form (Eq. 31) converge to a solution $x^*$ of $min_x \|Ax - b\|_M$ if and only if

$$0 < \varepsilon \leq \lambda_k \leq 2/\mu(T A^T M A) - \varepsilon \tag{32}$$

where $\varepsilon$ is an arbitrary small but fixed constant and $\mu(.)$ is the spectral radius (the largest positive eigenvalue). If in addition $x^0 \in \mathcal{R}(T A^T)$ then $x^*$ is the unique solution of minimum $T^{-1}$ - norm (minimum 2-norm if $T = I$).

For SART, Eq. (31) was written in the following matrix form:



$$x^{k+1} = x^k + \lambda_k D_r^{-1} A^T D_c^{-1}(b - A_x^k) \tag{33}$$

where the diagonal matrices $D_r$ and $D_c$ are defined in terms of the row and the column sums:

$$D_r = diag(\|a^i\|_1) \qquad\qquad D_c = diag(\|a_j\|_1) \tag{34}$$

We do not include weights in this method. The convergence for SART was independently established in (Eq. 31, 34), where it was shown that $\mu(D_r^{-1} A^T D_r^{-1} A) = 1$ and that convergence therefore is guaranteed for $0 < \lambda_k < 2$. Therefore, regarding the approach to constrain apriori condition, SWART uses a unit covariance matrix, and considering that at the top of the troposphere (10-15 km) there is no water vapour, the corresponding top voxels are forced to have the value of zero.

The apriori data can be used as the first guess of $x$, to speed up the convergence by reducing the number of the necessary iterations.

To define the number of iterations, the back projection technique was implemented as a stop criteria. Following Eq. (27), $A\,x^k = b^k$ is close to the experimental data $b = b^0$, i.e., $|b_0 - Ax^k| = min$.

Using the back projection technique where b represents the observations in Eq. (27). If we use the computed x to retrieve b ($b_{SWART}$) we can calculate residuals ($Res$) and use it for quality control (Eq. 35).

$$Res = b - b_{SWART} \tag{35}$$

**3.6 Overview of the 5 models and retrievals achieved**

This study uses 5 models with different inversion technique, i.e., SVD, Least-Square adjustment (LS) and SART. Retrievals
are obtained in 3D mainly and one model in 2D (TUO model), and these have been proceeded for $N_w$ and $\rho_{wv}$, except for TUW model ($N_w$ only) and UBI model ($\rho_{wv}$ only). Figure 5 summarises the characterisation of covariance operator of data and apriori model, and the quality check for tomographic retrievals.

| Tomography model | Inversion | Dim. | Retrievals | Covariance operator data | Covariance operator a priori model | Quality check |
|---|---|---|---|---|---|---|
| BIRA | SVD, weighted & damped LS adjustment | 3D | $\rho_{wv}$, $N_w$ | 10% | 90% | Resolution matrix, Covariance matrix, Spread |
| WUELS | Kalman filter with selective SVD | 3D | $\rho_{wv}$, $N_w$ | Diagonal obs. error fed | Diagonal height dependent | Condition number and variance – covariance $\rho_{wv}$, $N_w$ |
| TUW | TSVD, iterative & weighted LS adjustment | 3D | $N_w$ | Elevation dependent weighting | Altitude dependent weights | RMS of weighted residuals |
| UBI | SART | 3D | $\rho_{wv}$ | Unit covariance matrix | Unit covariance matrix | Condition number and convergence |
| TUO | LS adjustment | 2D | $\rho_{wv}$, $N_w$ | / | / | / |

**Figure 5: Overview of the 5 models and retrievals achieved**





## 4 Sensitivity tests and methodological improvements

### 4.1 Presentation of the sensitivity tests

The inverse problem treated by GPS tomography is ill-posed due to non-homogenous distribution of slants observations through the 3D grid (lack and redundancy), but this is also ill-conditioned due to the high number of parameter physically

embedded. This explains why the stabilisation of the tomographic solution remains a challenging task, although of methodological improvements (Rohm et al., 2014). In this study, software based on different techniques (SVD, damped least-squares adjustment; ART, Kalman filter) are considered to test improvements of the stabilisation of solutions. The first test concerns the behaviour of retrievals related to apriori condition applied. The second test looks at the improvement of the geometrical distribution (using stacking data and pseudo-slant observations). In addition, in a third test, the impact of the

observations uncertainty on the quality of tomographic results has been studied. Table 2 shows an overview of all tested model settings.

**Table 2: Steps studied, divided into data type, apriori data inclusion, stacking observation and impact of uncertainty on solutions.**

| Data type | Initial observations | | Pseudo-observations | | Observations with uncertainty | | | |
|---|---|---|---|---|---|---|---|---|
| Apriori<br>Stacking | in first epoch | every 6 h | in first epoch | every 6 h | in first epoch | | every 6 h | |
| | | | | | Max | Min | Max | Min |
| No | step 2,3 | step 1,4 | step 10b | step 10a | step 14b+ | step 14b- | step 14a+ | step 14a- |
| 30 min | step 9b | step 9a | step 11b | step 11a | step 15b+ | step 15b- | step 15a+ | step 15a- |
| 1 h | step 8b | step 8a | step 12b | step 12a | step 16b+ | step 16b- | step 16a+ | step 16a- |
| 2 h | step 7b | step 7a | step 13b | step 13a | step 17b+ | step 17b- | step 17a+ | step 17a- |

For all steps presented in Tab. 2, spatial and temporal distributions of the $N_w$ or $\rho_{wv}$ parameters have been determined. Interval calculation for all the steps in Tab. 2 is 30 minutes (except for step 1 and 2 which is every 6 hours). Columns "Max" and "Min" are related to steps 14 to 17 where an impact of uncertainty on solutions was tested. Label "Max" means a case with maximum uncertainty and label "Min" with minimum uncertainty.

Arrows in red in Fig. 6 indicate apriori condition from ACCESS-A, and in green from the previous tomography retrieval (TR). Steps 1 – 4 were designed to test apriori condition and time resolution of tomography models, whereas steps 7 – 9 were meant to test improvement of stacked data on solutions (using 2h-, 1h- to 30 minutes-stacked data respectively). Note that steps 5 and 6, missing in Tab. 2, with 3h- and 4h-stacked data, were initially planned; finally, we considered these 2 steps not relevant for improving nowcasting. Steps 10 – 13 were set up to test the improvement of pseudo-slants solutions,

whereas steps 14 – 17 were designed to test the impact of uncertainty on solutions. Figure 6 shows a simplistic illustration of the configuration used in steps 1 – 4. Note that steps (7b, 8b, 9b, 10b, 11b, 12b, 13b, 14b+, 14b-, 15b+, 15b-, 16b+, 16b-,



17b+, 17b-) used the same configuration as step 3, and steps (7a, 8a, 9a, 10a, 11a, 12a, 13a, 14a+, 14a-, 15a+, 15a-, 16a+, 16a-, 17a+, 17a-) the same as step 4.

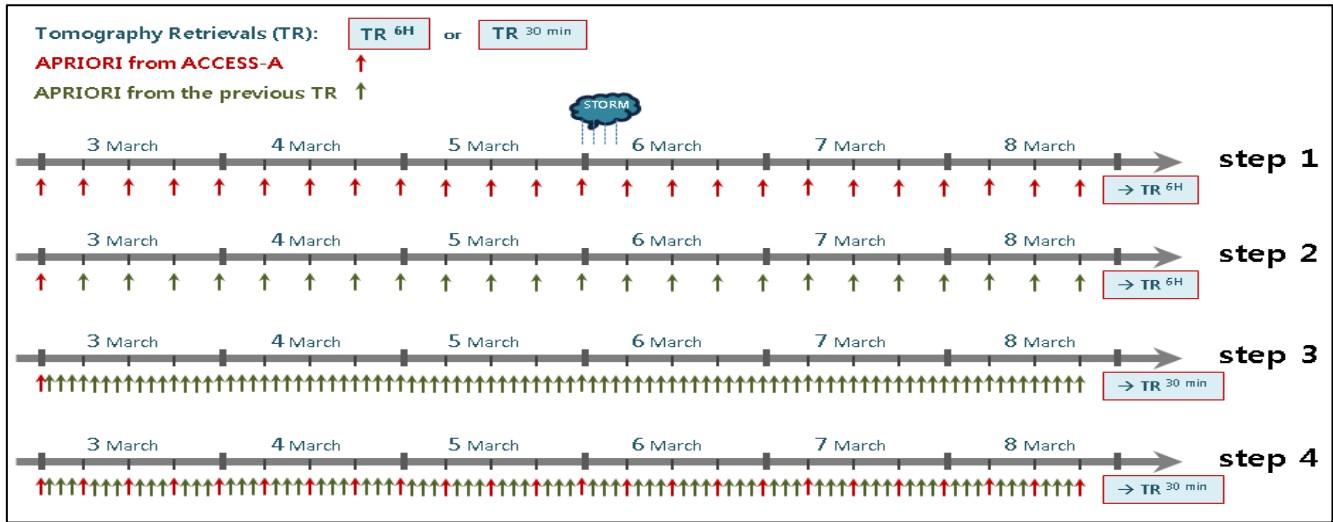

**Figure 6: Simplistic illustration of the configuration of steps 1-4 with time-location of the storm that occurred close to Melbourne.**

An increased geometrical distribution can be a way to obtain a better stabilisation of tomography solution. More voxels being crossed by slant GPS observations (SLANT$_{GPS}$) can improve the understanding of this meteorological situation. But this means that more inputs are considered in tomography processing (the mean number of SLANT$_{GPS}$ is provided for each epoch of calculation). Table 3 shows the status of the geometrical distribution (% of voxels crossed by SLANT$_{GPS}$ for the different steps considered), inside the inner tomography grid. Note that the mean Time of Processing is also indicated (a processor 2.66 GHz with 24 GB RAM has been used), as being a key parameter for nowcasting (non-linear quasi-quadratic dependency with the number of inputs). According to the distribution of rays from GPS stations inside the tomographic grid, for the 288 times of measurements from 3 to 10 March 2010, the mean percentage of voxels retrieved by our classic tomography models is 67.8% for the whole tomographic grid (geometrical distribution obtained without stacking and pseudo-slant observations). Using stacked slant data (30 min-, 1h- or 2h-stacking), an increase of the geometrical distribution (+1.5 %, +1.7 % or +4.2 %, respectively) and a consequent increase of the time of processing (×2, ×4 or ×16, respectively) have been observed. Mainly the layers between 2 and 9 km show a high increase of geometrical distribution (about +10%). Using pseudo-slant observations, an increase of +25 % is observed for all layers on average, with an increase of +30 % for the layers between 0 and 4 km, and an increase of +20 % between 4 and 9 km. The geometrical distribution is only increased by +3 % for the top layer, between 9 and 15 km (for more details about pseudo-slants, see section 4.3.2).



**Table 3: Geometrical distribution and time processing of our sensitivity tests.**

| *Type of tomography calculation* | | *No stacking* | *Stacked data (30 min)* | *Stacked data (1 h)* | *Stacked data (2h)* | *Pseudo-slant observations* | |
|---|---|---|---|---|---|---|---|
| | | | | | | *No stacking* | *Stacked data (30 min)* |
| *Mean number of* $SLANT_{GPS}$ | | 685 | 1370 | 2050 | 3400 | 3140 | 6275 |
| *Geometrical distribution (% of voxels crossed)* *for different layers* | *0-1 km* | 51.2 | 51.2 | 51.2 | 51.4 | 82.9 | 84.6 |
| | *1-2 km* | 58.3 | 58.3 | 58.3 | 61.6 | 88.0 | 89.4 |
| | *2-4 km* | 65.6 | 66.0 | 67.4 | 72.2 | 93.1 | 93.1 |
| | *4-6 km* | 73.6 | 75.0 | 76.4 | 82.6 | 95.8 | 97.2 |
| | *6-9 km* | 77.0 | 81.3 | 84.0 | 87.5 | 92.4 | 93.1 |
| | *9-15 km* | 61.8 | 67.4 | 67.8 | 70.6 | 64.6 | 68.0 |
| | *all* | 67.8 | 69.3 | 69.5 | 72.0 | 92.8 | 93.7 |
| *Indicator of mean time of processing (minutes)* | | 0.5 | 1 | 2 | 8 | 7 | 130 |

To visualise the results and validate our methodological improvements, tomography retrievals are compared with external observations (profiles of $N_w$ and $\rho_{wv}$ from RS and RO). We also look at the horizontal and vertical extension of our

tomography derived $N_w/\rho_{wv}$ structures and compare that with information from OMI and GOME-2. Profiles of $N_w$ and $\rho_{wv}$ from ACCESS-A have also been considered (as an example of forecasts), and being the apriori conditions used in our tomography calculations. ERA-Interim (Dee et al., 2011) profiles have also been prepared and compared.

Three various comparisons are shown within this study: GPS tomography versus RS profiles, GPS tomography versus RO

profiles and individual GPS tomography solutions to each other or to ACCESS-A and ERA-Interim fields. Since all mentioned techniques sense the tropospheric water vapour in a different way, also presented comparisons are based on different approaches. To compare RS profiles with GPS tomography results, or with ACCESS-A and ERA-Interim fields, RS profiles were linearly interpolated to individual heights of the tomography grid. Tomography retrievals and ACCESS-A/ERA-Interim estimates were then bilinearly interpolated to positions of RS stations – RS profiles are assumed to be

vertical (wind field not considered with the assumption that tomography outputs and RS profiles are simultaneous). Slant RO profiles were derived at heights of the tomography grid. Then, tomography retrievals and ACCESS-A/ERA-Interim estimates were bilinearly interpolated to latitudes/longitudes of RO profiles. For comparisons between two individual GPS tomography solutions or between a GPS tomography solution and ACCESS-A fields no interpolation had to be applied and a direct confrontation in 3D tomographic voxels was realised.





## 4.2 Results about the impact of apriori condition and time-evolution of tomography retrievals

**Table 4: Sensitivity of $N_w$ tomography retrievals to apriori / time resolution, normalised RMS**

| Normalised RMS | Compared to | ACCESS-A | ERA-Interim | Tomo. BIRA | | | Tomo. WUELS | | | Tomo. TUW | | | Tomo. TUO | |
|---|---|---|---|---|---|---|---|---|---|---|---|---|---|---|
| | | | | 1 | 2 | 3 | 1 | 2 | 3 | 1 | 2 | 3 | 1 | 2 |
| RS | 0.2 to 1.5 km | 0.16 | 0.21 | 0.18 | 0.20 | 0.22 | 0.20 | 0.24 | 0.26 | 0.17 | 0.20 | 0.22 | 0.21 | 0.56 |
| | 1.5 to 4 km | 0.15 | 0.19 | 0.22 | 0.36 | 0.38 | 0.28 | 0.43 | 0.35 | 0.21 | 0.29 | 0.30 | 0.20 | 0.61 |
| | 4 to 8 km | 0.50 | 0.67 | 0.74 | 3.20 | 2.96 | 2.36 | 2.26 | 2.91 | 1.00 | 4.16 | 3.23 | 0.74 | 17.08 |
| | 8 to 13 km | 0.98 | 0.97 | 2.33 | 10.94 | 2.12 | 73.57 | 83.05 | 37.23 | 8.80 | 30.90 | 19.15 | 1.50 | 311.49 |
| | 0 to 8 km | 0.25 | 0.34 | 0.36 | 1.26 | 1.14 | 2.11 | 3.64 | 1.23 | 0.72 | 2.22 | 1.82 | 0.35 | 32.57 |
| | 0 to 10 km | 0.25 | 0.38 | 0.35 | 1.12 | 1.06 | 0.85 | 0.88 | 1.05 | 0.42 | 1.36 | 1.11 | 0.36 | 5.30 |
| | All layers | 0.34 | 0.41 | 0.60 | 2.34 | 1.19 | 9.94 | 11.15 | 5.57 | 1.47 | 5.06 | 3.36 | 0.50 | 43.57 |

To look at the impact of apriori condition on tomography retrievals, this section will look at step 1, 2 and 3. RS are launched at 00:00 and 12:00 UTC, so the results of step1 and 4 are the same. In Table 4 values of normalised RMS are given which are also used in most of the following tables with results presentation. Normalised RMS values were computed from relative differences of two corresponding values from two solutions. Relative difference mean that an absolute difference of two

values is divided by one of the values. This way of statistics computation allows a reasonable comparison of studied $N_w$ and $\rho_{wv}$ parameters, whose absolute values change significantly with height. Table 4 shows that in step 1, wet refractivity estimates from GPS tomography models are usually the best in the bottom part of the troposphere (below 1.5 km), with two out of three models with comparable performance (less than 10 % degradation) than ACCESS-A, namely BIRA and TUW. Tomography retrievals (step 1 BIRA, WUELS, TUW) in the bottom part of the model are showing a better

agreement to RS than to ERA-Interim results, and TUO is of similar performance. However, it seems that this positive impact of GPS observations on the estimated troposphere state is diminished once the apriori data introduced to the tomography model are no longer fed in every epoch (step 2) or the time resolution of estimates is increased to 1 hour (step 3). In the second investigated layer (1.5 − 4 km) the tomography retrieved refractivity is usually of the same or slightly worse accuracy than the NWP-based solutions: BIRA and TUW by 0.02 (step 1). Once the apriori data are not fed into the

solution, the quality of retrievals drops substantially, by almost 40 %. The retrievals in the higher layers: 4 − 8 km and 8 − 15 km, for all models in this study are substantially worse than the ACCESS-A retrieval in all three steps. It has to be also noted




that TUO retrieval even though, less accurate than ACCESS-A, provide similar solution quality as ERA-Interim for all 10 km (0.36 to 0.38).

Similar pattern is shown for models producing water vapour density. Comparing to radiosonde in Melbourne, the tomography solution (step 1) is of similar quality to the ACCESS-A model in the first two layers of atmosphere (below 4
km), especially for models BIRA and UBI, while WUELS and TUO show a bit lower performance. Between 4 − 8 km only BIRA and TUO retrievals are of useful accuracy while WUELS and UBI show 5  and 4 times lower performance , respectively. Interestingly, the 8 − 13 km retrieval from TUO shows a better performance than ACCESS-A data.  Step 2 and 3 show a much worse solution especially in the higher levels of the model. BIRA model in step 1 reaches a better agreement with RS than ERA-Interim results in the whole 0 − 8 km profile.

**4.3 Results about the improvement of geometrical distribution**

Two types of tests to investigate improvement of the geometrical distribution of tomography retrievals have been proceeded: stacking of slant GPS data and additional pseudo-slant observations adding.

**4.3.1 Interest of data stacking for mid-troposphere retrievals**

Stacking of GPS data (SWD/SIWV) has been tested for 3 stacking periods (30 minutes, 1 hour and 2 hours before the time
of tomography reconstruction). Therefore, the two processing strategies, as illustrated in Fig.6 as  step 3 and 4 (respectively for steps 9b, 8b, 7b and steps 9a, 8a, 7a) have been appled. The observation stacking procedure, in theory should decrease condition number of design matrix (Eq. 10, 20) by improving the observation geometry, however it might also decrease accuracy of retrieval if the troposphere is active and changes rapidly within the stacking window.

The result for steps 9b, 8b and 7b were compared to RS and to step 3 as this solution was also initiated with apriori data from NWP only in the first epoch. Models WUELS and UBI in steps 9b, 8b and 7b perform very similar to step 3 as long as the troposphere below 4 km is considered, in the layer 4 − 8 km, both deliver improved solution: WUELS shows in step 8b a 30 % improvement over step 3, and UBI performs in step 9b  about 20 % better than in step 3. However, both solutions are worse than ACCESS-A and ERA-Interim, especially above 8 km. Surprisingly TUO does not improve results with stacking
of more observations except for the layer 1.5 − 4 km. BIRA model produces somewhat different results, in a layer 0.2 − 1.5 km the solution for step 9b is better than in step 3 and in ERA-Interim. In all higher layers until 13 km step 7b is much better than step 3, while for the top most layer it attains better performance than ACCESS-A and ERA-Interim (0.90 vs 0.98 and 1.06, respectively). In case of TUW (not shown in Tab. 5 as this model produced $N_w$ only) the performance of the model is better in step 9b than in step 3, however above 8 km height, the best performance is found for the longest stacking period of
2hours (step 7b).





**Table 5: Sensitivity of $\rho_{wv}$ tomography retrievals to data stacking without apriori data, normalised RMS is shown.**

| Normalised RMS | Compared to | ACCESS-A | ERA-Interim | Tomo. BIRA | | | | Tomo. WUELS | | | | Tomo. UBI | | | | Tomo. VSB | | |
|---|---|---|---|---|---|---|---|---|---|---|---|---|---|---|---|---|---|---|
| | | | | 3 | 9b | 8b | 7b | 3 | 9b | 8b | 7b | 3 | 9b | 8b | 7b | 9b | 8b | 7b |
| RS | 0.2 to 1.5 km | 0.16 | 0.21 | 0.20 | 0.19 | 0.20 | 0.38 | 0.25 | 0.25 | 0.28 | 0.28 | 0.19 | 0.19 | 0.19 | 0.20 | 0.72 | 0.61 | 0.62 |
| | 1.5 to 4 km | 0.16 | 0.19 | 0.35 | 0.36 | 0.36 | 0.31 | 0.33 | 0.34 | 0.32 | 0.33 | 0.30 | 0.30 | 0.31 | 0.31 | 1.02 | 2.31 | 1.61 |
| | 4 to 8 km | 0.50 | 0.67 | 3.78 | 3.49 | 3.43 | 0.72 | 2.95 | 2.83 | 2.32 | 2.55 | 6.31 | 5.16 | 5.17 | 5.22 | 44.52 | 43.38 | 31.83 |
| | 8 to 13 km | 0.98 | 1.06 | 13.81 | 8.14 | 7.22 | 0.90 | 44.34 | 47.94 | 33.15 | 30.84 | 18.05 | 283.2 | 277.3 | 282.6 | 844.7 | 814.2 | 543.2 |
| | 0 to 8 km | 0.25 | 0.33 | 1.27 | 1.20 | 1.18 | 0.45 | 1.05 | 1.02 | 0.88 | 0.94 | 1.98 | 1.65 | 1.66 | 1.67 | 13.34 | 13.44 | 9.89 |
| | 0 to 10 km | 0.26 | 0.38 | 1.66 | 1.48 | 1.42 | 0.48 | 1.22 | 1.19 | 1.79 | 1.91 | 4.19 | 2.77 | 2.58 | 2.64 | 56.45 | 63.55 | 44.59 |
| | All layers | 0.34 | 0.42 | 2.84 | 2.06 | 1.93 | 0.51 | 6.46 | 6.88 | 4.91 | 4.68 | 3.99 | 36.84 | 36.11 | 36.79 | 117.3 | 113.5 | 76.56 |

In Tab. 6, the result for steps 9a, 8a and 7a are compared to RS and to step 4 as this solution was also initiated with apriori data every 6 hours. Individual solutions show somewhat different pattern to what was observed in previous case of steps 7b,

5 8b and 9b (Tab. 5). Models: BIRA and UBI show a similar performance for step 4 and 9a (shortest 30 min-stacking) and decreased accuracy for longer stacking period. WUELS reaches a better performance in step 8a (1 hour-stacking), while TUO is performing best in the first two layers (until 4 km) in step 9a and then in step 8a (layer $4 - 8$ km) to return to better performance in 9a step for the topmost layer. Both UBI and BIRA have comparable accuracy retrievals in the first two layers to ACCESS-A values and slightly better than ERA-Interim. In case of TUW (not shown in Tab. 6 as this model produced $N_w$

10 only) the performance of the model is similar in all steps.

**Table 6: Sensitivity of $\rho_{wv}$ tomography retrievals to data stacking with apriori data, normalised RMS.**

| Normalised RMS | Compared to | ACCESS-A | ERA-Interim | Tomo. BIRA | | | | Tomo. WUELS | | | | Tomo. UBI | | | | Tomo. VSB | | |
|---|---|---|---|---|---|---|---|---|---|---|---|---|---|---|---|---|---|---|
| | | | | 4 | 9a | 8a | 7a | 4 | 9a | 8a | 7a | 4 | 9a | 8a | 7a | 9a | 8a | 7a |
| RS | 0.2 to 1.5 km | 0.16 | 0.21 | 0.16 | 0.17 | 0.18 | 0.20 | 0.21 | 0.21 | 0.18 | 0.19 | 0.16 | 0.16 | 0.17 | 0.16 | 0.22 | 0.22 | 0.22 |
| | 1.5 to 4 km | 0.16 | 0.19 | 0.17 | 0.19 | 0.21 | 0.26 | 0.26 | 0.26 | 0.22 | 0.24 | 0.20 | 0.19 | 0.19 | 0.19 | 0.28 | 0.33 | 0.30 |
| | 4 to 8 km | 0.50 | 0.67 | 0.55 | 0.61 | 0.70 | 0.84 | 2.77 | 2.72 | 1.31 | 1.39 | 1.95 | 1.53 | 1.51 | 1.47 | 0.87 | 0.69 | 0.72 |
| | 8 to 13 km | 0.98 | 1.06 | 1.50 | 1.88 | 2.27 | 3.10 | 91.51 | 98.76 | 135.6 | 116.3 | 20.63 | 156.6 | 148.9 | 136.6 | 1.18 | 1.28 | 1.47 |
| | 0 to 8 km | 0.25 | 0.33 | 0.28 | 0.30 | 0.34 | 0.40 | 0.96 | 0.94 | 0.52 | 0.55 | 0.69 | 0.57 | 0.56 | 0.55 | 0.43 | 0.39 | 0.39 |
| | 0 to 10 km | 0.26 | 0.38 | 0.28 | 0.30 | 0.34 | 0.41 | 1.78 | 1.73 | 1.18 | 1.12 | 3.33 | 2.65 | 2.54 | 2.20 | 0.43 | 0.39 | 0.38 |
| | All layers | 0.34 | 0.42 | 0.43 | 0.50 | 0.58 | 0.74 | 12.28 | 13.17 | 17.40 | 15.02 | 3.18 | 20.07 | 19.11 | 17.55 | 0.52 | 0.50 | 0.53 |





### 4.3.2 Interest of pseudo-slant observations for low- and mid-troposphere retrievals

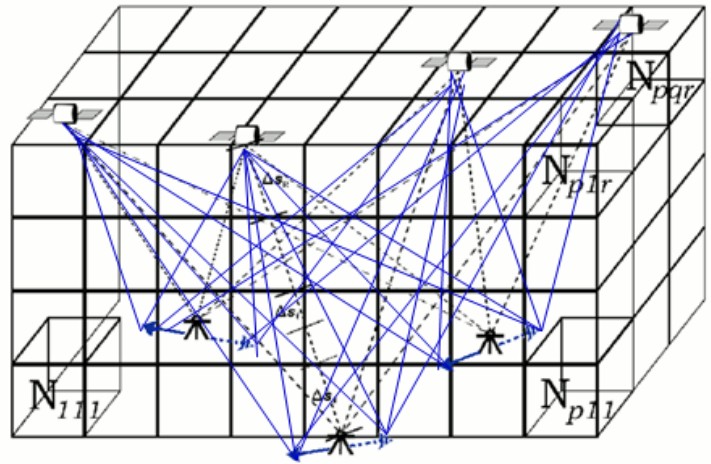

**Figure 7: Illustration of additional pseudo-slant observations in GPS tomography. The dashed lines show slant observations. The blue and dashed-blue arrows represent respectively the direction of horizontal gradients and their opposites (for a defined distance of e.g. 20 km) that are used to generate pseudo-slants.**

This section investigates the interest of adding pseudo-slants observations (SLANT$_{GPS}$) in tomography retrievals. The pseudo-slants have been implemented according to the orientation of delay gradient (Brenot et al., 2013). To obtain homogeneous repartition of sites (based on CORS network), a distance of 20 km on both side of a station has been considered, in the direction defined by GPS gradient. This means that for 2 pseudo-sites, the wet delay and the water vapour content have been propagated at a 20-km distance using the amplitude of the gradient, and the pseudo-SLANT$_{GPS}$ in direction of GPS satellites are estimated using isotropic mapping function (GMF in this case; see Boehm et al., 2006) and considered in tomography calculations, as illustrated in Fig. 7. Note also that 2 other pseudo-stations located at 10 km of the station have been used, considering gradient orientation and its opposite direction. This makes a total of 4 pseudo-sites with 4 sets of additional SLANT$_{GPS}$. As shown in Tab. 3, the use of pseudo-slants increases the geometrical distribution by +30 % and +20 % for the layers between $0 - 4$ km and $4 - 9$ km respectively. To test the implication of improving the geometry with pseudo-slant observations, step 4 (with a regular initialisation from NWP outputs; see Fig. 6) and step 3 (only previous tomography retrievals used as apriori, making a 'stand-alone solution') are compared with steps 10a and 10b respectively; see Table 7. Steps 11a and 11b (with 30 min data stacking) are also shown. Steps 12a, 12b, and 13a, 13b with respectively 1h- and 2h-stacked pseudo-slants as originally planned, were not processed (steps 13a and 13b have been totally skipped) since they are not relevant for nowcasting. Steps 12a and 12b are not shown in Tab. 7 but the improvement of their geometrical distribution is small (see Tab. 3) and the tomography retrievals are equivalent for WUELS and UBI models (not calculated by BIRA model). The comportment of the 3 models considered are different. While WUELS improves in step 10a and 11a (over step 4), a degradation is observed in step 10b and 11b (over step 3). This conclusion is mainly driven by





analysis of layer 8 – 13 km. Even with an improvement, results are still far from RS data (normalised RMS of about 73 for WUELS, and about 1 for ACCESS-A and ERA-Interim). For UBI model, the same tropospheric layer drives the result, but with a degradation according to both steps 3 and 4. For BIRA model, tomography retrievals using pseudo-SLANT$_{GPS}$ with a regular initialisation from NWP show a small degradation from a normalised RMS of 0.43 to 0.87 and 1.27 for step 10a and

11a, respectively (mainly driven by high troposphere layer). However, the use of pseudo-SLANT$_{GPS}$ with 'stand-alone initialisation' (overall normalised RMS of 2.84 for step 3) obtains a consequent improvement with an overall RMS of 0.56 and 1.05 for steps 10b and 11b, respectively. The improvement takes place in all layers. In layer 8 – 13 km, the BIRA model shows even a slightly better performance than ACCESS-A (RMS of 0.97 for BIRA over 0.98 for ACCESS-A).

**Table 7: Sensitivity of $\rho_{wv}$ tomography retrievals with additional pseudo-slants, normalised RMS.**

| Compared to Normalised RMS | | ACCESS-A | ERA-Interim | Tomo. BIRA | | | | | | Tomo. WUELS | | | | | | Tomo. UBI | | | | | |
|---|---|---|---|---|---|---|---|---|---|---|---|---|---|---|---|---|---|---|---|---|---|
| | | | | 4 | 10a | 11a | 3 | 10b | 11b | 4 | 10a | 11a | 3 | 10b | 11b | 4 | 10a | 11a | 3 | 10b | 11b |
| RS | 0.2 to 1.5 km | 0.16 | 0.21 | 0.16 | 0.21 | 0.27 | 0.20 | 0.49 | 1.46 | 0.21 | 0.31 | 0.31 | 0.25 | 0.34 | 0.35 | 0.19 | 0.16 | 0.16 | 0.16 | 0.19 | 0.19 |
| | 1.5 to 4 km | 0.16 | 0.19 | 0.17 | 0.28 | 0.41 | 0.35 | 0.36 | 0.73 | 0.26 | 0.29 | 0.29 | 0.33 | 0.35 | 0.35 | 0.30 | 0.20 | 0.19 | 0.20 | 0.34 | 0.33 |
| | 4 to 8 km | 0.50 | 0.67 | 0.55 | 0.88 | 1.26 | 3.78 | 0.70 | 0.96 | 2.77 | 2.85 | 2.85 | 2.95 | 3.11 | 2.91 | 6.31 | 1.90 | 1.59 | 1.95 | 4.87 | 4.53 |
| | 8 to 13 km | 0.98 | 1.06 | 1.50 | 3.94 | 5.95 | 13.81 | 0.97 | 0.98 | 91.51 | 73.00 | 73.00 | 44.34 | 115.7 | 106.1 | 18.05 | 229.0 | 186.0 | 20.63 | 293.6 | 300.9 |
| | 0 to 8 km | 0.25 | 0.33 | 0.28 | 0.43 | 0.60 | 1.27 | 0.51 | 1.06 | 0.96 | 1.03 | 1.03 | 1.05 | 1.13 | 1.08 | 1.98 | 0.67 | 0.58 | 0.69 | 1.58 | 1.48 |
| | 0 to 10 km | 0.26 | 0.38 | 0.28 | 0.44 | 0.63 | 1.66 | 0.54 | 1.06 | 1.78 | 1.82 | 1.82 | 1.22 | 1.22 | 1.13 | 4.19 | 3.04 | 2.63 | 3.33 | 2.52 | 2.32 |
| | All layers | 0.34 | 0.42 | 0.43 | 0.87 | 1.27 | 2.84 | 0.56 | 1.05 | 12.28 | 10.03 | 10.03 | 6.46 | 15.45 | 14.21 | 3.99 | 29.21 | 23.76 | 3.18 | 38.08 | 38.91 |

## 4.4 Results about the impact on the precision of slant retrievals

To investigate the impact on the precision of slant retrievals, the uncertainties of parameters acting for obtaining SWDs and SIWVs have been considered. These parameters are the specific molar gas constant for dry air ($R_d$), the specific molar gas

constant for water vapour ($R_w$), the refractivity coefficients ($k_1$, $k_2$, $k_3$, and $k_2' = k_2 - k_1.R_d/R_w$), the surface pressure ($P_S$), the mean temperature of the column of air ($T_m$), the gravity in the centre of the atmospheric column pressure ($g_m$), the GPS observation of ZTD, the estimation of ZHD, ZWD, factor κ and IWV. More details about how these physical parameters are linked to wet delay and integrated water vapour content are presented by Brenot et al. (2006). Then finally, two kinds of empirical mapping functions (NMF of Niell, 1996, and GMF of Boehm et al., 2006) have been considered to estimate the

impact on SLANT$_{GPS}$. The impact on the precision of slant retrievals is presented by an absolute uncertainty (in the unit of tested parameter) and by a relative error (in %).



**Table 8: Absolute uncertainty and relative error of parameters related to GPS tomography.**

| Parameter | Value | Unit | Absolute uncertainty | Relative error |
|---|---|---|---|---|
| $R_d$ | 287.0586 | J/(kmol K) | $\pm$ 0.0055 | $\pm$ 0.002 % |
| $R_w$ | 461.525 | J/(kmol K) | $\pm$ 0.013 | $\pm$ 0.003 % |
| $k_1$ | 77.60 | K/hPa | $\pm$ 0.05 | $\pm$ 0.064 % |
| $k_2$ | 70.4 | K/hPa | $\pm$ 2.2 | $\pm$ 3.125 % |
| $k_3$ | 373900 | K²/hPa | $\pm$ 1200 | $\pm$ 0.321 % |
| $k_2$' | 22.1345 | K/hPa | $\pm$ 2.2352 | $\pm$ 10.090 % |
| **Parameter** | **Typical value** | **Unit** | **Absolute uncertainty** | **Relative error** |
| $P_S$ | 1000 | hPa | $\pm$ 2 ($\pm$ 1) | $\pm$ 0.200 % ($\pm$ 0.100 %) |
| $T_m$ | 285 | K | $\pm$ 1 ($\pm$ 0.5) | $\pm$ 0.351 % ($\pm$ 0.175 %) |
| $g_m$ | 9.807 | m.s$^{-2}$ | $\pm$ 0.022 | $\pm$ 0.227 % |
| ZTD | 2.54 | m | $\pm$ Formal error + 0.010 (+ 0.005) | $\pm$ 0.497 % ($\pm$ 0.296 %) |
| ZHD | 2.29 | m | $\pm$ 0.011 ($\pm$ 0.007) | $\pm$ 0.494 % ($\pm$ 0.326 %) |
| ZWD | 0.25 | m | $\pm$ 0.019 ($\pm$ 0.010) | $\pm$ 8.605 % ($\pm$ 4.503 %) |
| $\kappa$ | 159 | kg.m$^{-3}$ | $\pm$ 1.335 ($\pm$ 1.053) | $\pm$ 0.840 % ($\pm$ 0.622 %) |
| IWV | 40 | kg.m$^{-2}$ | $\pm$ 3.375 ($\pm$ 2.513) | $\pm$ 8.440 % ($\pm$ 6.260 %) |
| $L_{sym}^{wet}$ | variable$^*$, mean: 0.765 | m | variable$^*$, mean: $\pm$ 0.069 | mean: $\pm$ 9.020 % |
| $L_{az}^{wet}$ & ($G_{EW}$, GNS) | variable$^*$, mean: 0.027 [$G_{EW}$, $G_{NS}$] = [5, 5] | m [mm, mm] | variable$^*$, mean: $\pm$ 0.003 ($\pm$ 0.001) $\pm$ Formal error + [0.8, 0.6] ([0.4, 0.2]) | mean: $\pm$ 10.249 % ($\pm$ 5.137 %) |
| SWD | variable$^*$, mean: 0.792 | m | variable$^*$, mean: $\pm$ 0.072 ($\pm$ 0.039) | mean: $\pm$ 9.091 % ($\pm$ 4.924 %) |
| SIWV | variable$^*$, mean: 122 | kg.m$^{-2}$ | variable$^*$, mean: $\pm$ 12.230 ($\pm$ 6.238) | mean: $\pm$ 10.003 % ($\pm$ 5.098 %) |

$^*$depends on the elevation of GPS satellite and data considered

In the upper six lines of Tab. 8, we can see that the parameter with the highest relative error is the refractivity coefficient k'$_2$. The uncertainties of refractivity coefficients (k$_1$, k$_2$, k$_3$) provided by Bevis et al. (1994) have been used, and the combination of higher and lower values for these coefficients show an absolute error of about 10% for k'$_2$. The bottom part of Tab. 8 presents absolute uncertainty and relative error for typical values of the estimated parameters. In these columns, two types of uncertainty have been considered: one conservative (high values) and one more realistic (lower values). The results related to

more realistic absolute uncertainty and relative error are presented in brackets (e.g. conservative uncertainty of P$_S$ of 2 hPa against a more realistic uncertainty of 1 hPa). To estimate the uncertainty of g$_m$, the bias between the formulation of Saastamoinen (1972) and the formulation of Vedel et al. (2001) has been considered (relative error of about 0.2%). The relative error in ZTD measurements and in P$_S$, T$_m$ and ZHD estimates, are low (< 0.5%). On the other hand, the relative errors of ZWD and IWV are high: about 8.5% for both (conservative error) and 4.5% and 6.2% respectively for the more

realistic error. The analytic expression of the conversion factor κ, takes into account k$_3$, k'$_2$ and R$_d$ coefficients and T$_m$ parameters (Askne and Nordius, 1987). T$_m$ has been estimated using output from ACCESS-A model, with an absolute uncertainty of 1 K (or 0.5 K). A moderate relative error is found for κ (< 1%). Note this parameter can obtain higher relative error in case of severe weather condition and the occurrence of hydrometeors in the path travel of GNSS signal (Brenot et





al., 2006). The last parameter that we have considered is the uncertainty of the mapping function. The uncertainty of $SLANT_{GPS}$ has been estimated using the satellite visible from one station during one epoch of measurements (10 satellites). To test the uncertainty of SWDs, the uncertainty of $L_{sym}^{wet}$ of Eq. (2) has been considered using the absolute error of NMF and GMF mapping function (Niell, 1996; Boehm et al., 2006), as shown in Eq. (36):

$$L_{sym}^{wet,MIN}(\epsilon) = ZWD \cdot (GMF - |NMF - GMF|)$$

$$L_{sym}^{wet,MAX}(\epsilon) = ZWD \cdot (GMF + |NMF - GMF|)$$

(36)

where $GMF = mf_{sym}^{wet,GMF}(\epsilon)$ and $NMF = mf_{sym}^{wet,NMF}(\epsilon)$. This shows a relative error of 9 %. To obtain the uncertainty of $L_{az}^{wet}$ of Eq. (3), the formal error of gradients provided by geodetic software and the error estimated by Brenot et al. (2014)

have been considered. This brings a relative error of 10 % (5 %, more realistic). Finally, the uncertainty of SWD and SIWV, is respectively 9 % and 10 %, with more realistic errors of 5 % for both.

Table 8 shows the impact of the uncertainties, added to the $SLANT_{GPS}$ retrievals, on the estimation of $N_w$ and $\rho_{wv}$. The higher and lower estimates of slants have been used to look at the repercussion on tomography solutions. Table 9 summarises the results observed for 8 tests about the impact of uncertainty (steps 14a+ and 14a-, for higher and lower slants

respectively, without stacking; step 15a+ and 15a- with 30-min stacking; step 16a+ and 16a- with 1-hour stacking; step 17a+ and 17a- with 2-hours stacking). These steps used apriori from ACCESS-A every 6 hours and can be compared to steps 4, 9a, 8a and 7a. Thereby, the statistical results using more realistic uncertainty are presented in brackets (like in Tab. 8).

**Table 9: Sensitivity of $\rho_{wv}$ tomography retrievals to apriori / time resolution, normalised RMS.**

| Normalised RMS (Compared to) | | ACCESS-A | ERA-Interim | Tomo. BIRA | | | | | | | | | | | |
|---|---|---|---|---|---|---|---|---|---|---|---|---|---|---|---|
| | | | | 4 | 14a+ | 14a- | 9a | 15a+ | 15a- | 8a | 16a+ | 16a- | 7a | 17a+ | 17a- |
| RS | 0.2 to 1.5 km | 0.16 | 0.21 | 0.16 | 0.17 (0.17) | 0.16 (0.16) | 0.17 | 0.19 (0.18) | 0.15 (0.16) | 0.18 | 0.21 (0.20) | 0.15 (0.16) | 0.20 | 0.25 (0.22) | 0.14 (0.17) |
| | 1.5 to 4 km | 0.16 | 0.19 | 0.17 | 0.18 (0.18) | 0.17 (0.17) | 0.19 | 0.21 (0.20) | 0.18 (0.18) | 0.21 | 0.25 (0.23) | 0.19 (0.20) | 0.26 | 0.33 (0.29) | 0.23 (0.23) |
| | 4 to 8 km | 0.50 | 0.67 | 0.55 | 0.60 (0.58) | 0.50 (0.53) | 0.61 | 0.72 (0.67) | 0.48 (0.55) | 0.70 | 0.88 (0.79) | 0.48 (0.59) | 0.84 | 1.15 (1.00) | 0.51 (0.67) |
| | 8 to 13 km | 0.98 | 1.06 | 1.50 | 1.67 (1.59) | 1.28 (1.40) | 1.88 | 2.26 (2.08) | 1.43 (1.67) | 2.27 | 2.87 (2.58) | 1.58 (1.93) | 3.10 | 4.10 (3.62) | 2.01 (2.55) |
| | 0 to 8 km | 0.25 | 0.33 | 0.28 | 0.30 (0.29) | 0.26 (0.27) | 0.30 | 0.35 (0.33) | 0.25 (0.28) | 0.34 | 0.41 (0.38) | 0.26 (0.30) | 0.40 | 0.54 (0.47) | 0.28 (0.33) |
| | 0 to 10 km | 0.26 | 0.38 | 0.28 | 0.30 (0.29) | 0.26 (0.27) | 0.30 | 0.35 (0.33) | 0.26 (0.28) | 0.34 | 0.42 (0.38) | 0.27 (0.30) | 0.41 | 0.55 (0.48) | 0.30 (0.34) |
| | All layers | 0.34 | 0.42 | 0.43 | 0.47 (0.45) | 0.39 (0.41) | 0.50 | 0.59 (0.55) | 0.40 (0.45) | 0.58 | 0.72 (0.65) | 0.42 (0.50) | 0.74 | 0.98 (0.86) | 0.49 (0.61) |



We find consistent comportment of tomography retrievals according to the type of uncertainty (regular with strong impact, more realistic with less impact). We also observe that stacked data amplify the impact of uncertainty in tomography retrievals. Considering RS as a reference, the sensitivity of the bias between tomography and RS water vapour density for different kinds of slants (maximum/minimum uncertainty with regular/more realistic errors), in comparison to the bias

between NWP with RS, shows interesting results about the quality and possible improvements in SLANT$_{GPS}$ and about the validation of NWP model using tomography retrievals. Adding uncertainty measure to the observation matrix increases the accuracy of stochastic modelling for tomography retrievals and using apriori every 6h (in Tab. 10), stacked solution (30min - 15a-, 1h – 16a-, 2h – 17a-) shows a better performance in bottom 1.5 km. In the upper layer (1.5 – 4 km), solution is of lower quality than ACCESS-A but still better than ERA-Interim. Layer 4 – 8 km is again better resolved using BIRA model in

steps 15a- and 16a- (30min and 1h, respectively). The topmost layer is of lower quality in all steps. The same kinds of tests have been proceeded using apriori from ACCESS-A only for the first epoch (to be compared with steps 3, 9b, 8b and 7b). In comparison to the results of Tab. 10, the impact of the uncertainty is amplified due to the stand-alone situation of the tomography technique which is less constrained by the apriori condition and ACCESS-A model. To illustrate this, we can look at the layer (4 – 8 km) and the comparison of $\rho_{wv}$ from RS with estimates from ACCESS-A, ERA-Interim, step 9a, step

15a+ and step 15a-. That is shown in Tab. 10 with respective normalised RMS of 0.50, 0.67, 0.61, 0.72 (0.67), 0.48 (0.55); and also look at the comparison with step 9b, step 15b+ and step 15b- showing respective normalised RMS of 3.49, 3.50 (3.17) and 2.11 (2.49). Results in brackets correspond to more realistic uncertainty applied for steps 15a+, 15a-, 15b+ and 15b-.

## 5 Inter-comparison with Cross-validation of tomography models

**5.1 All steps – statistics**

Tomography models produce parameters that are usually of similar or worse quality than ACCESS-A retrievals (see Tab. 4, 5, 6, 7 and 9, and Fig. 8) with few exceptions i.e.: in the bottom part of the troposphere for BIRA and TUW tomography models (0.2 – 1.5 km) for which wet refractivity retrievals are of better quality comparing to ACCESS-A for tomography models fed with 6h apriori data, 30 minutes stacking and limited observation uncertainty (step 15a-). Moreover, BIRA

solution is only 10 % worse than ACCESS-A in all layers below 10 km height (step 1). Time stacking is improving the solution with respect to non-stacked one in all investigated cases for cases without continues feed of apriori data (steps 7b, 8b, 9b). For the lowest layer a short stacking (30min, step 9b) gives a better response in BIRA model, while for all other layers longer stacking works better (2hours step 7b), lowest discrepancy between tomography model and RS was found in layer 8 – 13 km and stacking 7b with (8 % improvement for BIRA solution). The use of pseudo-slant observations shows

variable results. Sometimes it doesn't bring any visible improvement in terms of RMS of solution with respect to the ACCESS-A accuracy (note that if the tomography models are of same quality as ACCESS-A in the RS location across





model domain then in other locations, without impact of radiosonde assimilated in the ACCESS-A, the data could be of better performance). But in some cases (e.g. layer 0 – 10 km), the use of pseudo-SLANT$_{GPS}$ with 'stand-alone initialisation' obtains a consequent improvement (see steps 10b and 11b compared to step 3 in Tab. 7). BIRA solution of step 10b is even 1% better than ACCESS-A for the layer 8 – 13 km.

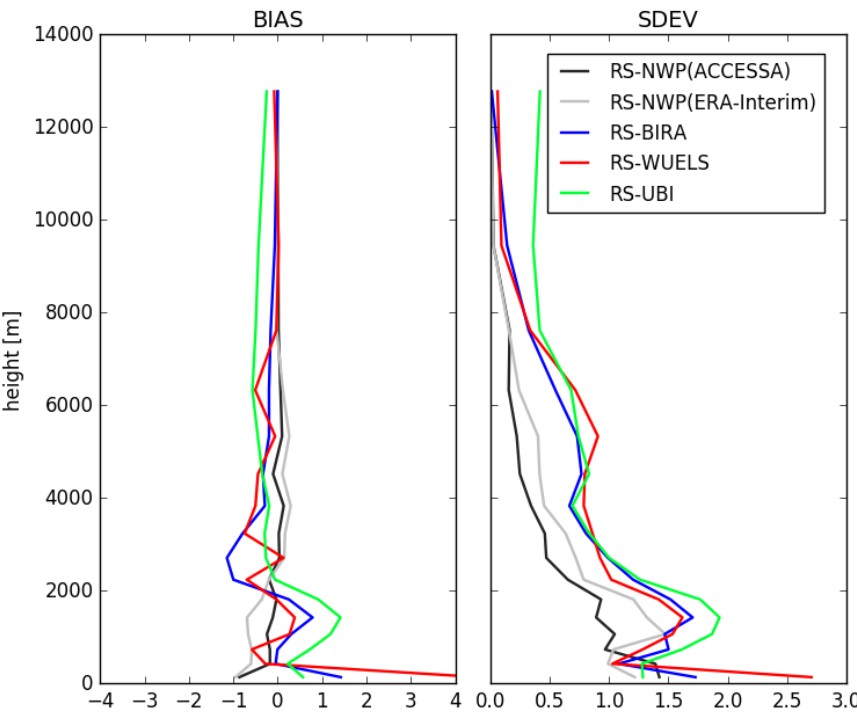

**Figure 8: Comparison of $\rho_{wv}$ retrieval from NWP models (ACCESS-A and ERA-Interim) and tomography models (BIRA, UBI, WUELS; step 9b) with respect to Melbourne radiosonde (RS) over whole 5 days period.**

Figure 8 illustrates the comportments of $\rho_{wv}$ retrievals from 3 tomography models (BIRA, UBI and WUESLS) with respect

10   to Melbourne RS estimates during whole 5 days period; comportments of NWP (ACCESS-A or ERA-Interim) are also presented. Even ACCESS-A and ERA-Interim show $\rho_{wv}$ estimates closer to RS measurements than tomography models, we can notice that the 3 tomography models show the same $\rho_{wv}$ – pattern in the layer 0.2 – 4 km. A trend of moistening is observed in the low troposphere by GPS tomography and is not seen by RS and NWP. Such similarity bring into the light the use of ensemble tomography solution, which could be a way to identify anomaly in the low troposphere, with respect to

15   NWP or other measurement technique. Note that tomography outputs and RS are assumed to be simultaneous (without considering the wind field that effected the trajectory of RS profiles). These assumptions can be the reason of the difference between tomography models and RS observations. In future work, we plan to consider wind speed in our comparison with RS, and to take a look to such an approach with ensemble tomography, especially for nowcasting application. A first investigation is shown in conclusion of this study.



## 5.2 Comparison to radio occultation

From the 3$^{rd}$ to the 8$^{th}$ of March 2010, there were 4 radio occultation (RO) events reported by CDAAC that coincide with the location of our tomography network. The $N_w$ – profiles from RO were compared to the BIRA, WUELS and TUW model

5    results, whereas the BIRA, WUELS, and UBI models were compared to $\rho_{wv}$ – profiles obtained from RO (wetPrf product). The profiles collected at 2010/03/03 08:07UTC, 2010/03/08 05:36UTC, 2010/03/08 07:33UTC (consult Figure 1), show no significant difference between RO, tomography retrievals, ACCESS-A and ERA-Interim results and follows a typical exponential pattern. Therefore, these profiles will not be discussed further.

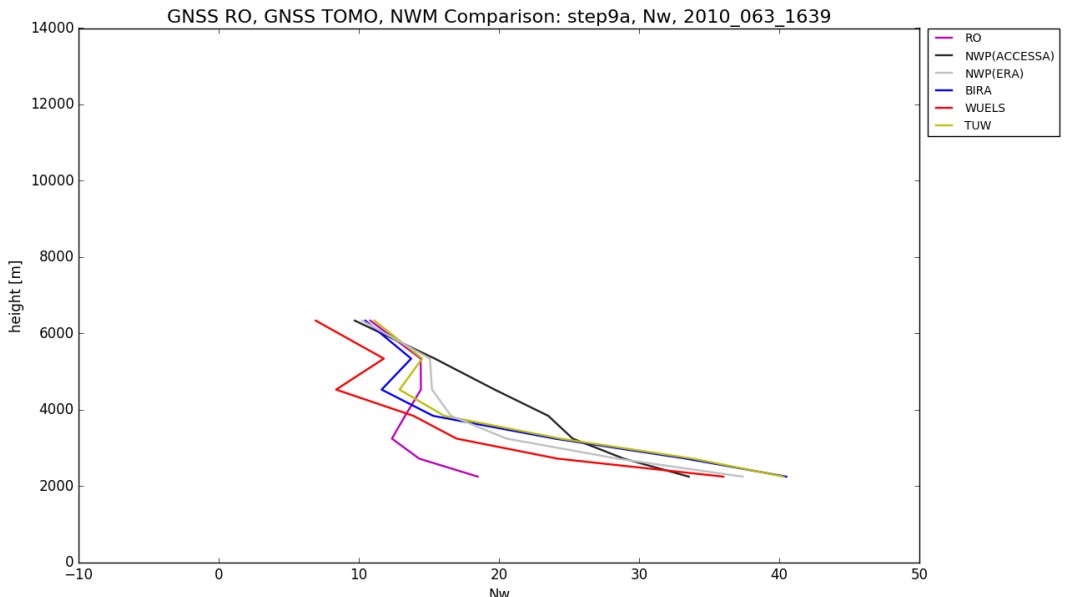

**Fig. 9: Profiles of $N_w$ distribution determined on the basis of Radio Occultation technique (purple), ACCESS-A (blue) service and three tomography models: BIRA (green), TOMO2 (red) and TUW (yellow)**

The Figure 9 presents profiles which run from south-west towards the north-east, over Australian Alps (see Fig. 1) and

15    shows results from the 4$^{th}$ march at 04:39PM UTC (08:39AM local time). Distribution of $N_w$ and $\rho_{wv}$ were calculated using initial observations in 30 minutes of calculation interval, without stacking and taking into account the apriori data by 6 hours. The results of the WUELS, BIRA and TUW models are similar, distinguishing two inversions in vertical profiles at heights between 4 and 6 km (that we can define as $N_w$ – anomaly). For the RO profile, there are also local minima at these heights, however, the largest inversion occurs below 4 km. The RO – and tomography – based values of $N_w$ and $\rho_{wv}$ significantly

20    differ from the ACCESS-A results in the lowest layer as later does not show any inversion below 4 km. All tomography-based profiles show a similar inversion layer to the one represented by RO profile, however the location of that layer is lower





in RO (from 2 km to 5 km) and higher in tomography profiles (from 3.8 km to 5.5 km). More importantly, the ERA-Interim result for this particular location is much closer to the tomography solution (BIRA, WUELS) and RO retrieval, showing a slight inversion between 5 and 6 km. The $N_w$ – anomaly retrieved by both tomography models is a relevant information, bringing motivation for using ensemble tomography solutions in future investigations.

5  **5.3 Selected results – only selected part of tomography network in the East and during selected epochs**

Based on the previous investigated RO profile, we decided to verify the performance of the tomography models with respect to ERA-Interim in the eastern part of this model (orange/red pixels with high top cloud detected by GOME-2 in the Fig. 1a). The investigation is focussed on the most accurate processing step 9a and times 12 and 18 UTC on DOY 063 (see Tables 10 and 11).

**Table 10: Normalised RMS of $N_w$ tomography retrievals in comparison to NWP (ACCESS-A, ERA-Interim), DOY 063 12 and 18 UTC, results from 9 voxels in the south-eastern part of network.**

| Solution | Height | | | | | | |
|---|---|---|---|---|---|---|---|
| | 0.2 to 1.5 km | 1.5 to 4 km | 4 to 8 km | 8 to 13 km | 0 to 8 km | 0 to 10 km | All layers |
| ACCESS-A - ERA | 0.28 | 0.21 | 0.20 | 0.45 | 0.23 | 0.23 | 0.26 |
| Step9a | | | | | | | |
| ACCESS-A – BIRA | 0.03 | 0.07 | 0.16 | 0.47 | 0.08 | 0.10 | 0.13 |
| ACCESS-A – WUELS | 0.49 | 0.52 | 0.50 | 76.29 | 0.50 | 0.52 | 9.98 |
| ACCESS-A – TUW | 0.04 | 0.06 | 0.11 | 3.96 | 0.07 | 0.08 | 0.55 |
| ERA – BIRA | 0.20 | 0.30 | 0.21 | 2.14 | 0.24 | 0.24 | 0.48 |
| ERA – WUELS | 0.35 | 0.78 | 0.52 | 53.40 | 0.55 | 0.57 | 7.16 |
| ERA – TUW | 0.20 | 0.30 | 0.19 | 3.69 | 0.23 | 0.24 | 0.67 |

15  **Table 11: Normalised RMS in % of $N_w$ tomography retrievals in comparison to NWP (ACCESS-A, ERA-Interim), DOY 063 12 and 18 UTC, results from all voxels.**

| Solution | Height | | | | | | |
|---|---|---|---|---|---|---|---|
| | 0.2 to 1.5 km | 1.5 to 4 km | 4 to 8 km | 8 to 13 km | 0 to 8 km | 0 to 10 km | All layers |
| ACCESS-A - ERA | 0.27 | 0.34 | 0.20 | 0.58 | 0.28 | 0.29 | 0.32 |
| Step9a | | | | | | | |
| ACCESS-A – BIRA | 0.08 | 0.11 | 0.20 | 0.65 | 0.13 | 0.15 | 0.19 |
| ACCESS-A - WUELS | 0.35 | 0.42 | 0.44 | 53.17 | 0.40 | 0.45 | 7.00 |
| ACCESS-A - TUW | 0.08 | 0.10 | 0.16 | 5.55 | 0.11 | 0.14 | 0.79 |
| ERA – BIRA | 0.18 | 0.24 | 0.21 | 2.62 | 0.21 | 0.22 | 0.51 |
| ERA - WUELS | 0.26 | 0.41 | 0.48 | 35.64 | 0.38 | 0.41 | 4.79 |
| ERA - TUW | 0.17 | 0.23 | 0.19 | 4.73 | 0.20 | 0.21 | 0.77 |



First of all, it is clear that in the first 10 km there is a small discrepancy between ACCESS-A and BIRA and TUW tomography models of 0.10 and 0.08 respectively. At the same time, there is a much larger discrepancy between ACCESS-A and ERA-Interim in the same height range (0.23). Taking into account this disagreement, it has to also be observed that the discrepancy between ERA-Interim – BIRA and ERA-Interim – TUW is smaller (0.20) in the range 0.2 – 1.5 km of height

than respective discrepancies in the ACCESS-A – ERA-Interim comparison (0.28). For the layer 4 – 8 km, the differences are not so distinct but still exists for model TUW (0.19) versus (0.20) in the ACCESS-A – ERA-Interim couple. The 1.5 – 4 km layer in tomography models is comparing less effectively to the ERA solution.

Table 11 shows the normalised RMS for all voxels in the same time as in Tab. 10, but for the whole tomographic network. ACCESS-A – ERA comparison clearly shows similar performance for bottom layer and for layer 4 – 8 km, whereas layer

1.5 – 4 km shows much worse quality (0.21 – selected voxels against 0.34 – all voxels). In the same time the discrepancy between ACCESS-A and TUW and BIRA models increased for all voxels with respect to selected voxels but decreased for ERA – BIRA and ERA – TUW models. This substantial improvement of tomography model solution is best visible in the 1.5 – 4 km band, where ACCESS-A – ERA couple shows 0.34 discrepancy and ERA – BIRA and ERA – TUW 0.24 and 0.23, respectively, it is also smaller than for selected 9 voxels solution (in Tab. 10) by 0.10.

**6 Meteorological interest and conclusion about future applications**

In this study we tested a number of variants for which the impact on the quality of tomographic results has been assessed. These variants are notably the apriori condition considered in tomography process, the quality of initial observations and the impact of observations uncertainty, the interest of stacking data and the use of pseudo-slant observations. We used five tomographic models: BIRA, WUELS, TUO, UBI and TUW; processing results were compared with data obtained from

ACCESS-A and ERA-Interim outputs, RS profiles and RO products.

Our sensitivity tests show that using 30-min stacking data, an improvement is obtained, without bringing too old data into tomography process, allowing a proper understanding of the meteorological situation. However, the increase of the geometrical distribution (Tab. 3) is weak using only 30-min stacking data. We find that the use of pseudo-slants can significantly improve the geometrical matrix of tomography retrievals (showing a better special representativeness) and can

bring, in some cases, an improvement. The tests of apriori condition of tomography show that the best results (with respect to RS) are obtained using the best apriori. This is why, assuming that ACCESS-A apriori was reasonably good (and close to RS profiles), tomography retrievals, using regular apriori from NWP, show the best results. It looks that the "stand-alone" strategy (using the previous tomography retrievals as the apriori of the next calculation) is not always successful. A divergence in results is obtained; especially when the apriori is far to the real state of the atmosphere. In future work, we plan

to test "stand-alone" solution in a sliding window mode: i.e. for example, having an apriori at ($t_0$ – 3h), and then to consider the tomography solution at $t_0$ using the "stand-alone" solution.





The results about the quality of initial observations and the impact of observations uncertainty, is the following. Even the relative error of parameters (linked to tomography retrievals) which characterise the moistening (e.g. refractivity coefficient $k_2'$, SWD or SIWV), is about one order of amplitude higher than for the others, the mean global impact on GPS tomography for all the layer is moderated (from 4% without stacking, to 24% with 2h stacking), as shown in Tab. 9. However, the

interesting result of this sensitivity test is that, for some cases with lower (or higher) values of SWD or SIWV, closer tomography estimates are obtained in comparison to RS profiles. With the use of strong and more realistic uncertainties, respectively applied positively and negatively, this is finally 4 sets of SLANT$_{GPS}$ that we have tested to assess the sensitivity of tomography retrievals. The order of magnitude in observations is transferred to tomography retrievals (e.g. in Tab. 9, an increase in observations amplitudes shows an increase in normalised RMS from tomography with respect to RS, meaning

that lower SLANT$_{GPS}$ applied in tomography show closer results to RS).

The most relevant result obtained in this study is that moistening anomaly can be found in tomography retrievals compared to NWP forecast. This is all the more interesting if such anomaly is identified by several tomography models, as shown in Fig. 9. We plan in the future to investigate the interest of ensemble tomography solution (from several models) in nowcasting system. An illustration of this potential application is shown in Fig. 10 and 11. During this severe weather event,

a typical configuration with dry/moist air contrast initiated deep convection (Choy et al., 2011). In the night of the 5[th] of March 2010, lifting of moist was coming from the southeast side of Melbourne. Figure 10 shows the comparison of the $\rho_{wv}$ horizontal sections from ACCESS-A and BIRA tomography solution. The bubble of water vapour (located south-east of Melbourne) is clearly visible in GPS tomography (for the horizontal section at 0.738 km in Fig. 10, and in the vertical section for a fixed longitude of 144.575°E where high values of $\rho_{wv}$ are observed around the latitude of -38°N). The

convective cell occurring very strong precipitation over Melbourne city from midnight to 04:00 UTC on 6 March 2010, is located south-east of Melbourne at 06:00 UTC. However we can see convective cells on the north and east side of Melbourne (see radar precipitation in Fig. 10). No dry/moist contrast is observed in the vertical sections obtained from ACCESS-A outputs (left images in Fig. 11). The forecast of precipitation was not successful. On the other hand, if we look at the vertical sections obtained from GPS retrievals, strong dry/moist contrasts are observed (between the latitudes -38°N

and -37°N, top right image in Fig. 11, and between the longitudes 143°E and 145°E, bottom right image).





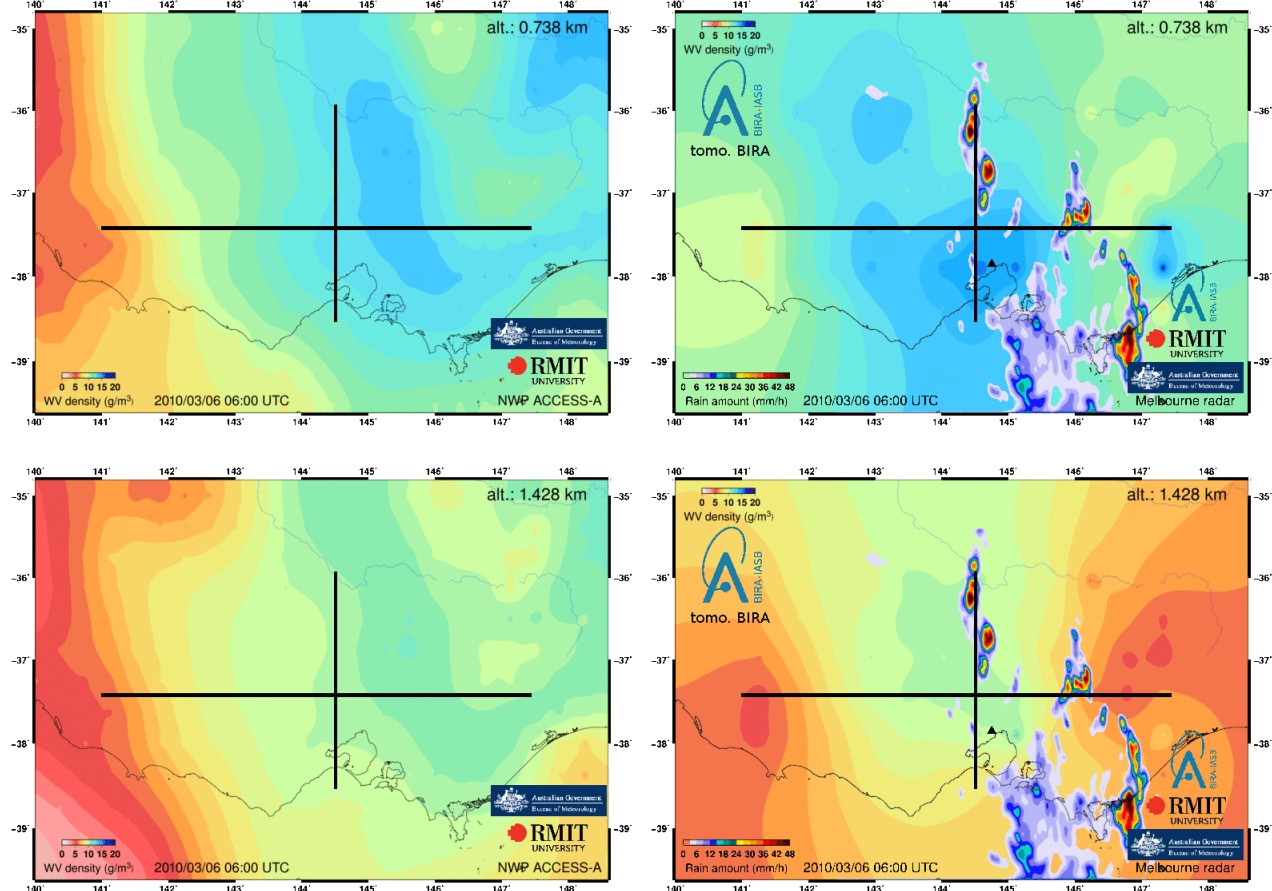

**Fig. 10: Horizontal section of $\rho_{wv}$ from ACCESS-A NWP (left side), and from BIRA tomography solutions (right side).**

**Radar precipitation has been superimposed on tomography section. Black lines shows the location of vertical sections of Fig. 11.**

**Upper images show sections at the altitude of 0.738 km and on the bottom is shown sections at 1.428 km.**





**Fig. 11: Vertical section of $\rho_{wv}$ from ACCESS-A NWP (left side), and from BIRA tomography solutions (right side).**

**Upper images show sections for fixed longitude of 144.575°E, and on the bottom is shown sections for a fixed latitude of -37.433°N.**



Finally to conclude, even we try to use independent external observations from RS and RO to assess the impact of several tests, more work is still needed to define precisely the best strategy to be used in GPS tomography. For example, the use of a dense network of external observations (RS or other profiler techniques) could be very useful to validate an optimal GPS tomography. Another way to validate these tests can be to use synthetic tomography (SLANT$_{GPS}$ simulations in NWP applied to assess synthetic tomography retrievals). It is important to notice that this study does not contain an evaluation of SWDs and SIWVs quality, which were computed from ZTDs and horizontal gradients estimated from GPS observations and used in GPS tomography as the main input data. Some level of imperfection in tomography results can come from this side. Moreover, post-fit residuals cleaned from systematic effects which were not used in this study could be beneficial for GPS GPS slant observations under severe weather conditions. Note also that the hypothesis of straight ray propagating has been considered in this study for modelling path delay from GPS station to ground-based receiver in our tomography models. Because the bending effect is negligible over 10° (Elgered et al., 1991), the inversion problem becomes linear and can be formulated using the discrete theory. However, the bending effect which can affect elevation under 10° (Zus et al., 2012, Möller and Landskron, 2018) has not been considered in our study. This needs to be considered in future work.

**Acknowledgement:**

This work has been achieved in the frame of the European COST Action ES1206 GNSS4SWEC (GNSS for Severe Weather and Climate monitoring; http://www.cost.eu/COST_Actions/essem/ES1206) aiming for studying the use of GNSS tropospheric products for high resolution NWP and sever weather forecasting (http://gnss4swec.knmi.nl/wg2). This investigation is also a contribution to the International Association of Geodesy (http://www.iag-aig.org) and the Solar-Terrestrial Centre of Excellence (http://www.stce.be).

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
