# Peer review of "Cross-validation of GPS tomography models and methodological improvements using CORS network"

_Atmospheric Measurement Techniques, 2018_

## Referee Comment (RC1) · Anonymous Referee #2 · 7 Dec 2018

This paper analyzes a number of variants which influence the quality of tomographic result using five models. Radiosonde and GNSS RO data are also used to validate the tomographic result. Generally speaking, this paper is well-organized and may have some reference significance, but still has some places requires clarified. Based on this, I recommended this paper for a moderate revision.

Specific comments:

P1L28-30, I don't think this expression is proper. Yes, more observations can be obtained but with similar paths, which would increase the instability of the design matrix of tomography model.

[Figure]

P6L4, what's the elevation angle mask of GNSS observation?

P6L9-13, what're the accuracies of meteorological parameters (P and T) derived from ACCESS-A weather model? What kind of interpolated method is used to obtain the corresponding parameters for the location of GPS station? What about the accuracy of the interpolated meteorological data?

P6L32-33, why the one-way residual is not considered? How much does this term matter to the tomographic result?

P7L15, I think the superscript of a left part in Eq. (5) maybe hydrostatic, but I'm not sure.

P8L6, the author says that the radio-sounding balloon is quite expensive, can you give a general value?

P10L10, what's your principle to select the horizontal size of the voxels in the inner grid? In my opinion, the size may be relatively large.

P29L8, it should be better if the authors can provide the comparison of water vapor profiles between radiosonde and different tomography models at some specific epochs, e.g. before, occurrence and after the storm.

---

## Author Comment (AC1) · 16 Jan 2019

Sorry, it looks that the name of one of the author as not been properly written in the pdf manuscript. This is Damian Tondaś, not Damian Tondás

---

## Referee Comment (RC2) · Anonymous Referee #3 · 25 Feb 2019

General Comment: The manuscript presented the assessments of the tomographic water vapor fields solved from 5 models. Authors have tested the impacts of various factors including initial conditions, data stacking, pseudo-slant observations and the uncertainty of GPS observations on the tomographic solutions. This paper is well written and structured. I thus recommend it for publication in AMT after a moderate modification.

Specific comments: 1 It is mentioned in section 3 that the voxels of the outer grid are considered inside a band with a width of about 400 km. Since the water vapor is assumed to be homogeneous distribution within each voxel, the huge outer voxels may

bring nonnegligible errors to the tomographic solution. Please show more details on how to include the outer voxel in the tomography model?

2 Line 16 of page 22, 'appled' to 'applied'

3 Stacking data is a good way to improve the tomographic results. Is the time span of the stacked data same to the time resolution of the tomography? If no, please indicate clearly in the paper the concept of data stacking.

4 Since during extreme weather conditions, the air mass will change very quickly within 10 minutes or higher. It seems that a resolution of 30 min is used in the study. Is it possible to reconstruct the water vapor fields with an interval of 10 min or even higher?

5 Pseudoslant SWDs in direction of GPS satellites are estimated using isotropic mapping function. How did you get the ZWDs for each pseudo-site? If the anisotropic part is not considered in the calculation of pseudo SWDs, it will have the same effects to the use of horizontal constraint on the tomography. Do you think so?

---

## Author Comment (AC2) · 25 Mar 2019

The answer is in PDF format

[Figure]

**Author's Response to RC1 (7 December 2018):**

In this author's response, the text (normal style) answers point by point to the comment of the anonymous Referee #2 (text in bold). The text in blue correspond to author's change in the manuscript.

Before starting properly this answer to RC1, we would like to mention this 2 corrections that have been applied to this manuscript:

1) the affiliation of Riccardo Biondi has been changed to:

Dipartimento di Geoscienze, Università degli Studi di Padova, Padova, Italy

2) The error in the name of one of the author has been corrected: Damian Tondaś, not Damian Tondás

**RC1 from the anonymous Referee #2:**

**This paper analyzes a number of variants which influence the quality of tomographic result using five models. Radiosonde and GNSS RO data are also used to validate the tomographical result. Generally speaking, this paper is well-organized and may have some reference significance, but still has some places requires clarified. Based on this, I recommended this paper for a moderate revision.**

**Specific comments:**

**P1L28-30, I don't think this expression is proper. Yes, more observations can be obtained but with similar paths, which would increase the instability of the design matrix of tomography model.**

The expression you speak about is the following: "*However, the use of data stacking and pseudo-observations can significantly improve the quality of the retrievals, due to a better geometrical distribution and a better coverage of mid- low-troposphere parts.*"

I agree with the fact that the use of stacking (5 minutes) brings quite similar paths, but the use of pseudo-slants is different in term of paths, and especially in term of geometric matrix (lower troposphere. with a better scan and more voxels are crossed for other layers). For this reason, no modification of the text is applied. I also agree that stacking will have negative impact on the time resolution, which means that it is assumed that for all stacked observations refractivity is constant.

**Fig. 1.**

---

## Author Comment (AC3) · 25 Mar 2019

The answer is in pdf format
* * *
[Figure]

**Author's Response to RC2 (25 February 2019):**

In this author's response, the text (normal style) answers point by point to the comment of the anonymous Referee #3 (text in bold). The text in blue correspond to author's change in the manuscript.

**RC2 from the anonymous Referee #3:**

**General Comment: The manuscript presented the assessments of the tomographic water vapor fields solved from 5 models. Authors have tested the impacts of various factors including initial conditions, data stacking, pseudo-slant observations and the uncertainty of GPS observations on the tomographic solutions. This paper is well written and structured. I thus recommend it for publication in AMT after a moderate modification.**

**Specific comments: 1 It is mentioned in section 3 that the voxels of the outer grid are considered inside a band with a width of about 400 km. Since the water vapor is assumed to be homogeneous distribution within each voxel, the huge outer voxels may bring nonnegligible errors to the tomographic solution. Please show more details on how to include the outer voxel in the tomography model?**

The reason behind this outer grid, is to keep considering in our calculations slant$_{GNSS}$ of stations located close to the edge of the inner grid, and for which the ray does not reach the top of the tomographic grid inside the inner grid. With this respect this allows our tomography models to use more data than without using the outer grid. We are aware that the retrievals obtained for the outer grid are not high-quality estimates, and hence these values were not used for analysis presented in the manuscript. Moreover, the outer grid voxels were design to absorb (in a numeric sense) biases that would be introduced otherwise by not taking into account the part of the delay out of the model. Studies using outer and inner grid and only inner with removal of excessive path delay (Hanna et al., 2019) using raytracing shows very limited impact of the later solution. Therefore we don't think the use of outer grid bring error to our retrievals (especially because the error concern voxels for high troposphere with very low water vapour density).

Hanna, N., Trzcina, E., Möller, G., Rohm, W., and Weber, R.: Assimilation of GNSS tomography products into WRF using radio occultation data assimilation operator, Atmos. Meas. Tech. Discuss., https://doi.org/10.5194/amt-2018-419, in review, 2019.

**2 Line 16 of page 22, 'appled' to 'applied'**

Ok thank you this has been modified, L16P22: "applied"

**Fig. 1.**